# VLOD-TTA: Test-Time Adaptation of Vision-Language Object Detectors

## Abstract

Vision–language object detectors (VLODs) such as YOLO-World and Grounding DINO achieve impressive zero-shot recognition by aligning region proposals with text representations. However, their performance often degrades under domain shift. We introduce VLOD-TTA, a test-time adaptation (TTA) framework for VLODs that leverages dense proposal overlap and image-conditioned prompt scores. First, an IoU-weighted entropy objective is proposed that concentrates adaptation on spatially coherent proposal clusters and reduces confirmation bias from isolated boxes. Second, image-conditioned prompt selection is introduced, which ranks prompts by image-level compatibility and fuses the most informative prompts with the detector logits. Our benchmarking across diverse distribution shifts – including stylized domains, driving scenes, low-light conditions, and common corruptions – shows the effectiveness of our method on two state-of-the-art VLODs, YOLO-World and Grounding DINO, with consistent improvements over the zero-shot and TTA baselines. [1]

## 1 Introduction

Object detectors (ODs) localize and classify objects in images (Zou et al., 2023), with applications ranging from surveillance (Mishra & Saroha, 2016) and autonomous driving (Gupta et al., 2021) to augmented reality (Ghasemi et al., 2022) and medical imaging (Li et al., 2019). Recently, vision–language ODs (VLODs) such as YOLO-World (Cheng et al., 2024) and Grounding DINO (Liu et al., 2024) have demonstrated strong zero-shot (ZS) recognition. Pretrained on large corpora of image–text and region–text pairs (Shao et al., 2019), these models learn a shared visual–semantic space that aligns region-level visual features with textual representations. This alignment enables generalization beyond the supervised label set, allowing recognition of previously unseen categories at inference without additional training.

As with other vision-language models (e.g., CLIP (Radford et al., 2021)), VLODs—despite strong ZS capability—exhibit performance degradation under distribution shift between pretraining and test domains (Shu et al., 2022). Fine-tuning VLODs with millions of parameters on a large-scale dataset is computationally expensive and often compromises ZS performance. We therefore study test-time adaptation (TTA) for VLODs, where the model is adapted on-the-fly during inference using only unlabeled target data while preserving ZS capability.

Although no TTA method currently exists for VLODs, several approaches have been proposed for vision–language models (VLMs) in classification. They typically minimize marginal predictive entropy over augmented views (Shu et al., 2022). While standard entropy minimization represents a good baseline, it has two limitations for OD: (i) it amplifies confirmation bias by sharpening the highest class score, producing overconfident mislocalized proposals (Farina et al., 2024), and (ii) it ignores proposal structure, assigning the same weight to isolated or cross-instance boxes as to overlapping, mutually consistent clusters. As illustrated in Fig. 1, standard entropy minimization increases the scores equally for both *person* and *dog* classes without regard to spatial coherence, leading to a dog false positive.

Another common strategy to improve VLM robustness is prompt ensembling: averaging multiple prompt templates per class (Radford et al., 2021). However, for VLODs, we empirically find that

---

[1]The code will be made available after acceptance.

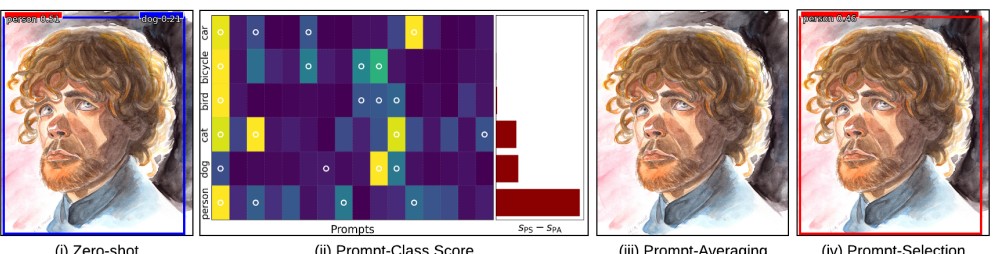

Figure 1: **Motivation (IWE). Left→right:** (i) proposals from two classes—**Person (red)** (cluster size = 167, max score = 0.14) and **Dog (blue)** (cluster size = 45, max score = 0.15); (ii) ZS scores fall below the threshold, resulting in a missed detection; (iii) standard entropy minimization over-confidently produces a dog false positive; and (iv) our IoU-weighted entropy minimization focuses updates on dense clusters and suppresses isolated boxes.

Figure 2: **Motivation (IPS). Left→right:** (i) ZS predictions with a correct detection **Person (red)** and a false positive **Dog (blue)**; (ii) prompt–class score heatmap with circles marking prompts selected by our image-conditioned strategy and right-margin bars showing $S_{\text{PS}} - S_{\text{PA}}$; (iii) prompt averaging (PA) reduces the class score, producing no detections; and (iv) prompt selection (PS) suppresses the dog false positive while preserving the person detection.

prompt averaging yields marginal gains and, in some cases, degrades performance. As shown in Fig. 2, prompt averaging reduces the class score for the person below the detection threshold, resulting in a missed detection.

To overcome these limitations, we propose **VLOD-TTA**, a TTA framework for VLODs with two components: **IoU-weighted entropy minimization (IWE)** and **Image-conditioned prompt selection (IPS)**. Modern ODs generate dense, overlapping proposals that act as free test-time augmentations, providing partially redundant views of the same instance. IWE exploits this spatial redundancy by assigning each proposal a weight proportional to its local IoU affinity with overlapping, class-consistent proposals. This focuses adaptation on consistent proposals and reduces confirmation bias from isolated or mislocalized proposals. In Fig. 1, IWE increases scores within the dominant *person* cluster rather than uniformly boosting all proposals.

Second, we introduce image-conditioned prompt selection (IPS). Instead of averaging all prompts, IPS selects image-relevant prompts for each class. Prompts are ranked by an image-level compatibility score defined as the mean class-specific logit across proposals, and we retain the top $\rho$ per class. Then, the retained prompt logits are combined with the OD logits. This preserves information from image-relevant prompts while disregarding irrelevant ones. As shown in Fig. 2, IPS selects suitable prompts and increases the person score relative to prompt averaging, leading to a correct detection.

**Our main contributions are summarized as follows:**

1. **TTA Framework for VLODs:** We introduce VLOD-TTA, to our knowledge, the first test-time adaptation framework for VLODs.

2. **IoU-weighted Entropy Minimization (IWE):** A detection-specific entropy objective is introduced that weights each proposal by its local same-class IoU affinity, concentrating adaptation on spatially coherent proposals and mitigating confirmation bias from mislocalized boxes.

3. **Image-conditioned Prompt Selection (IPS):** An IPS strategy is proposed that selects the most relevant prompts for a particular image while disregarding irrelevant ones.

4. **VLOD-TTA Benchmark:** We conduct extensive experiments to evaluate VLOD-TTA and benchmark common TTA techniques for VLODs on 6 mainstream detection datasets and 15 common corruptions, with a total of 96 distinct test scenarios. Results on two popular VLODs demonstrate the robustness and efficacy of our method.

## 2 RELATED WORK

**Test-Time Adaptation.** TTA improves robustness under domain shift by updating a small subset of parameters during inference using only test data. Wang et al. (2021) minimizes predictive entropy while updating batch-normalization parameters over batches of images. Zhang et al. (2022) replaces the batch requirement with multi-view augmentation and minimizes marginal entropy across multiple views, enabling single-sample adaptation. For VLMs, most TTA methods can be classified into two families: prompt-tuning and cache-based. Prompt-tuning optimizes a small set of continuous prompt tokens in the text encoder for each test image. Shu et al. (2022) updates soft prompts per image by minimizing marginal entropy across augmentations. Feng et al. (2023) improves over TPT by using diffusion-based augmentation, at a higher computational cost. Cache-based methods maintain a small test-time memory of high-confidence target features and their pseudo-labels. In Karmanov et al. (2024), a dynamic queue of recent feature–label pairs calibrates the current predictions through similarity-weighted aggregation without requiring backpropagation. Zhang et al. (2024) maintains image and text prototypes and optimizes small residual parameters on both modalities using entropy and contrastive objectives, aligning the two spaces. These methods focuses only on classification, where adaptation is performed at the image level and does not involve region proposals or a detection head, so localization is not addressed. In contrast, our VLOD-TTA adapts at the proposal level for open-vocabulary detection, improving both localization and classification.

**Vision-Language Object Detectors.** VLODs localize and recognize categories specified by text, relaxing the closed-set constraint of conventional ODs. Early transfer methods leverage contrastive image–text models such as CLIP (Radford et al., 2021) to supervise the OD's classifier head, enabling ZS recognition (Gu et al., 2022; Zhong et al., 2021; Zareian et al., 2021). Vocabulary scaling decouples localization from classification by training large-vocabulary classifiers on image-level labels while keeping proposals class-agnostic (Zhou et al., 2022). Grounded pretraining unifies detection and phrase grounding to learn language-aware object representations (Li et al., 2022). Grounding DINO (Liu et al., 2024) further fuses language with a transformer detector via language-conditioned queries and cross-modal decoding. On the efficiency side, YOLO-World (Cheng et al., 2024) introduces reparameterizable vision–language fusion for real-time open-vocabulary detection. Despite strong ZS results, VLOD performance drops significantly under domain shift. In our experiments, we evaluate YOLO-World (CNN-based) and Grounding DINO (transformer-based), demonstrating the effectiveness of VLOD-TTA across both architectures.

**Test-Time Adaptation for Object Detectors.** TTA enhances detectors' robustness against domain shifts without source data. Chen et al. (2023) formulates TTA-OD as mean-teacher self-training with feature-alignment regularization and performs multi-step adaptation with augmentations. Yoo et al. (2023) is architecture-agnostic, updates lightweight adapters, employs a stability-aware objective, and schedules updates. Cao et al. (2025) uses object-level contrastive alignment with class-wise confidence thresholds and selective parameter restoration to limit drift and forgetting. Ruan & Tang (2024) adapts from a single image and stabilizes pseudo-labels with an IoU-guided filter. Although these approaches report strong gains, many rely on heavy augmentations and multi-step updates, which are not ideal for real-world TTA. Moreover, these methods are tailored to closed-set ODs with a fixed, vision-only label space, where models are pre-trained on labeled source data that closely matches the target domain. In contrast, our VLOD-TTA adapts VLODs in an open-vocabulary label space, leveraging both visual and textual cues. For VLODs, Medeiros et al. (2025) is the only work that adapts across modalities, but in a supervised setting with labeled target data. To the best of our knowledge, *no prior work addresses TTA for VLODs*.

## 3 PROPOSED METHOD

The detailed architectural diagram of VLOD-TTA is shown in Fig. 3. Our method has two main components: IoU-weighted Entropy Minimization (IWE) and Image-Conditioned Prompt Selection

Figure 3: **Overview of our VLOD-TTA.** Given an input image and a set of prompts, the text encoder produces embeddings that interact with region proposals via the vision–language head to compute similarity scores. IPS performs top-$\rho$ prompt selection and averages the selected prompts to obtain per-proposal class scores. Then, it combines per-proposal entropy with IoU-based weights to form an IWE objective that drives robust TTA.

(IPS). IWE is responsible for weighting the proposal entropies by local IoU consistency to focus adaptation on reliable regions. Meanwhile, IPS ranks prompts by an image-level similarity score and retains the most informative prompts. In this section, we provide preliminaries and then describe each component in detail.

## 3.1 PRELIMINARY DEFINITIONS

**Vision–Language Object Detection.** A VLOD couples a visual detector with a text encoder operating in a shared embedding space. For zero-shot inference on an image $X \in \mathbb{R}^{C \times H \times W}$, the visual detector outputs $N$ candidate boxes $B = \{b_i\}_{i=1}^N$ and region features $\{\mathbf{v}_i\}_{i=1}^N$ with $\mathbf{v}_i \in \mathbb{R}^d$. For each category name $y_k$ in a label set $Y$ with $|Y| = K$, the text encoder produces an embedding $\mathbf{t}_k \in \mathbb{R}^d$. We compute similarity as:

$$s_{i,k} = \hat{\mathbf{v}}_i^\top \hat{\mathbf{t}}_k, \qquad \hat{\mathbf{v}}_i = \frac{\mathbf{v}_i}{\|\mathbf{v}_i\|_2}, \quad \hat{\mathbf{t}}_k = \frac{\mathbf{t}_k}{\|\mathbf{t}_k\|_2}. \tag{1}$$

The final detections are obtained by applying detector-specific post-processing, such as threshold filtering and non-maximum suppression.

**Entropy Minimization for OD.** Given the class scores $s_{i,k}$ for proposal $i$ and class $k$, the categorical posterior for proposal $b_i$ is $p_{i,k} = \left[\mathrm{softmax}(s_{i,1}, \ldots, s_{i,K})\right]_k$. The Shannon entropy (Shannon, 1948) per proposal is defined as $\mathcal{H}(\mathbf{p}_i) = -\sum_{k=1}^K p_{i,k} \log p_{i,k}$. TTA minimizes the empirical entropy over proposals in the image, which is given by:

$$\mathcal{L}_{\mathrm{Ent}} = \frac{1}{N} \sum_{i=1}^N \mathcal{H}(\mathbf{p}_i). \tag{2}$$

This objective sharpens each proposal's class posterior while reducing model uncertainty.

## 3.2 IOU-WEIGHTED ENTROPY MINIMIZATION (IWE)

VLODs produce large numbers of candidate boxes per image. In standard configurations, YOLO-World (YW) produces approximately $8,400$ proposals, and Grounding DINO (GD) approximately $900$. Post-processing, such as non-maximum suppression or top-$k$ filtering, removes most proposals, yet their spatial co-occurrence carries exploitable information. Regions with many mutually overlapping proposals that predict the same class are more likely to correspond to that class, whereas regions with few or dispersed proposals are less likely to be correct. A standard entropy objective that treats proposals independently ignores this structure and assigns the same weight to both cases, which can sharpen predictions in low-overlap regions and amplify confirmation bias.

VLOD-TTA integrates an IWE objective that allocates more weight to groups of mutually overlapping boxes and less to singletons, so adaptation is driven by regions with class-consistent overlap. Using the categorical posterior $\mathbf{p}_i$ and the entropy $\mathcal{H}(\mathbf{p}_i)$, let $\hat{c}_i = \arg\max_k p_{i,k}$ be the predicted class for proposal $b_i$. For each class $c$, construct a class-specific IoU graph $G_c = (V_c, E_c)$, whose vertices are the proposals with predicted label $c$. Two vertices $u$ and $v$ are connected by an edge in $E_c$ when $\text{IoU}(b_u, b_v) \geq \theta$, where $\theta \in [0, 1]$ is a fixed threshold. We define clusters as the connected components of $G_c$. A connected component is a maximal subset of $V_c$ in which any two vertices are linked by a path that uses only edges in $E_c$. Let $\mathcal{C}(i)$ denote the component that contains vertex $i$, and define the weight of proposal $b_i$ as $w_i = |\mathcal{C}(i)|^\gamma$, where $|\mathcal{C}(i)|$ is the number of proposals in that component and $\gamma \geq 0$ controls how strongly component size affects the objective. The IWE objective is defined as:

$$\mathcal{L}_{\text{IoU}-\text{Ent}} = \frac{\sum_{i=1}^{N} w_i \, \mathcal{H}(\mathbf{p}_i)}{\sum_{i=1}^{N} w_i}. \tag{3}$$

The weights depend only on the IoU graph and are treated as constants during backpropagation, so gradients flow through $\mathcal{H}(\mathbf{p}_i)$ only. This encourages updates in regions with many overlapping proposals that agree on the same class and reduces the influence of isolated or dispersed proposals.

## 3.3 Image-Conditioned Prompt Selection (IPS)

VLM performance is highly sensitive to the text prompt wording. Using multiple prompt forms yields an ensemble over text descriptions, and CLIP reports higher zero-shot accuracy when averaging text embeddings across multiple templates per class (Radford et al., 2021). We observe that this strategy is not effective for VLODs and can sometimes degrade performance. Rather than averaging across prompts uniformly, IPS is employed to select effective prompts per image and adapt the text representation at test time.

Suppose that for each class $k \in \{1, \ldots, K\}$ we have a pool of $T$ prompts $\{t_{k,1}, \ldots, t_{k,T}\}$ with embeddings $\{\mathbf{e}_{k,t} \in \mathbb{R}^d\}$. For proposals $\{b_i\}_{i=1}^{N}$ with region features $\{\mathbf{v}_i\}_{i=1}^{N}$, compute the similarities as $z_{i,k,t} = \hat{\mathbf{v}}_i^\top \hat{\mathbf{e}}_{k,t}$, where $\hat{\mathbf{v}}_i$ and $\hat{\mathbf{e}}_{k,t}$ are $\ell_2$-normalized vectors. The image-prompt similarity score is computed for each prompt using the empirical mean over proposals, defined as $r_{k,t} = \frac{1}{N} \sum_{i=1}^{N} z_{i,k,t}$. To suppress irrelevant prompts, for each class $k$, VLOD-TTA selects the top-$\rho$ fraction of prompts ranked by $\{r_{k,t}\}_t$ (see App. A.1 for details). Let $\mathcal{S}_k$ denote the indices of the top-$\rho$ prompts of $\{r_{k,t}\}$ for class $k$. VLOD-TTA then aggregates class scores for each proposal by averaging only the selected prompts as $\tilde{z}_{i,k} = \frac{1}{|\mathcal{S}_k|} \sum_{t \in \mathcal{S}_k} z_{i,k,t}$.

To further align text and region embeddings, a lightweight residual vector is introduced on the text side (Zhang et al., 2024). Let $\Delta \in \mathbb{R}^d$ be a learnable residual added to each prompt embedding, with:

$$\tilde{\mathbf{e}}_{k,t} = \frac{\mathbf{e}_{k,t} + \Delta}{\|\mathbf{e}_{k,t} + \Delta\|_2}. \tag{4}$$

Let $s_{i,k}$ denote the class score for the original OD, computed using Eq. (1). The final score is computed as:

$$g_{i,k} = \lambda \, \tilde{z}_{i,k} + (1 - \lambda) \, s_{i,k}, \qquad \lambda \in (0, 1). \tag{5}$$

## 3.4 Model Update

Although VLOD-TTA can be applied to any subset of parameters, we keep the base network fixed and adapt only a small set of adapter parameters (Houlsby et al., 2019; Chen et al., 2024). Adapters are lightweight and can be inserted independently of the underlying architecture. For YW, we insert adapters in the backbone and neck, while for GD, we insert adapters only into the text backbone. We attribute this difference to the architectural and pretraining differences between the two VLODs (see Sec. 4.3 for empirical validation). In YW, the detector and text encoder are treated as decoupled blocks and interact only through cosine similarity at the final scoring stage. Updating the text encoder therefore mainly changes classification scores and yields limited gains. This is consistent with the YW pre-training result that fine-tuning the text encoder can hurt performance (Cheng et al., 2024). We omit adapters in the head to reduce computational complexity, since adding head adapters did not improve performance in our ablations. GD instead uses early cross-modal fusion, where the

text encoder is tightly integrated into detection, so adapting the text encoder can directly influence both localization and classification. During pretraining, optimizing the text encoder jointly with the vision backbone results in better performance (Liu et al., 2024). GD tends to overfit when optimizing the vision encoder on a single image, due to its much heavier architecture (172M parameters for GD vs 13M for YW). Let $\Theta$ denote all network parameters and write $\Theta = (\Theta_{\text{frozen}}, \Phi, \Delta)$, where $\Phi$ denotes the adapter parameters, and $\Delta$ the residual parameter (defined in Eq. (4)). At test time, $(\Phi, \Delta)$ are zero-initialized as $(\Phi_0, \Delta_0)$ and the fused score $g_{i,k}$ is given by Eq. (5). To accelerate IoU graph construction, we retain the top-$M$ proposals ranked by $\max_k g_{i,k}$. The IWE optimizes $(\Phi, \Delta)$ in a single adaptation step. As the model is optimized on a single image, the adapted parameters are not guaranteed to generalize to other images, so $(\Phi, \Delta)$ are reset to $(\Phi_0, \Delta_0)$ after each prediction.

## 4 RESULTS AND DISCUSSION

This section reports an empirical evaluation of VLOD-TTA. We first describe the VLOD-TTA benchmark and present the main results, followed by a detailed analysis to elucidate the underlying mechanisms of VLOD-TTA. Finally, we provide ablation studies to assess VLOD-TTA robustness.

### 4.1 BENCHMARKING VLOD-TTA

We construct a benchmark that evaluates ZS performance, four TTA baselines adapted to VLODs, and our VLOD-TTA. It assesses four types of domain shift: texture/style, weather, illumination, and common corruptions, across seven datasets. We report results for YOLO-World and Grounding DINO using mean average precision (mAP) following the COCO protocol (Lin et al., 2014).

**Datasets.** **Watercolor/ClipArt/Comic:** Watercolor2k, ClipArt1k, and Comic2k (Inoue et al., 2018) are stylized artistic datasets used to evaluate robustness to synthetic/artistic style. **Cityscapes:** Cityscapes contains urban street scenes across multiple European cities (Cordts et al., 2016). We use it to study urban driving domain shift across geography, weather, and time of day. **BDD100K:** BDD100K is a large driving dataset spanning day and night, multiple cities, and diverse weather (Yu et al., 2020). We use it to assess real-world distribution shifts. **ExDark:** A low-light object detection dataset (Loh & Chan, 2019). It measures robustness under poor illumination. **PASCAL-C:** Michaelis et al. (2019) corrupts PASCAL-VOC (Everingham et al., 2010) with 15 common corruption types at 5 severities, following Hendrycks & Dietterich (2019), to evaluate OD robustness.

**Baselines.** **Zero-shot:** Pretrained VLODs are used for inference with no TTA (Liu et al., 2024; Cheng et al., 2024). **Test-Time Prompt Tuning (TPT):** We adapt TPT (Shu et al., 2022) from classification to OD by optimizing only the text prompt vectors at test time. Candidate boxes are selected using entropy, and a marginal-entropy objective is minimized on those proposals. **Visual Prompt Tuning (VPT):** Following the TPT pipeline, we optimize only visual prompts (Jia et al., 2022). Visual prompting is effective for modality adaptation (Medeiros et al., 2025). **DPE:** We adapt DPE (Zhang et al., 2024) to OD by maintaining per-class text/visual cache memory constructed from high-confidence proposals. At test time, only the residual cache parameters are adapted using a marginal-entropy objective combined with a cache-contrastive loss. **Adapter Tuning:** We adapt lightweight bottleneck adapters using the TENT objective (Wang et al., 2021). This removes dependence on specific model parameters and allows a fair comparison with our method.

**Implementation Details.** We use the official YW implementation (Cheng et al., 2024) and the MMDetection implementation of GD (Liu et al., 2024). Unless stated otherwise, YW-small and GD-Tiny are used. We report AP using the COCO API (Lin et al., 2014). Each experiment uses a batch size of 1 and a single adaptation step. We set $\gamma = 1.1$, $\rho = 0.25$, $M = 600$, and $\lambda = 0.3$ for YW and $\lambda = 0.1$ for GD. We use Conv-Adapters (Chen et al., 2024) in YW with reduction $= 4$ and kernel $= 3$, and MLP Adapter (Houlsby et al., 2019) in GD with reduction $r = 16$ (see App. A.4 for details). We use $T = 16$ GPT-generated prompts per class. For our experiments, we used ChatGPT-5 to generate textual prompts with the instruction: "Generate 16 prompts for each object category: $< category\ list >$." Text embeddings are computed offline before adaptation for YW and during adaptation for GD. For a fair comparison, we keep all other hyperparameters identical to the ZS baseline.

| | YOLO-World | | | | | | | | | | | | | | | | | |
| --- | --- | --- | --- | --- | --- | --- | --- | --- | --- | --- | --- | --- | --- | --- | --- | --- | --- |
| | Watercolor | | | ClipArt | | | Comic | | | Cityscapes | | | BDD100K | | | ExDark | | |
| Method | mAP | $AP_{50}$ | $AP_{75}$ | mAP | $AP_{50}$ | $AP_{75}$ | mAP | $AP_{50}$ | $AP_{75}$ | mAP | $AP_{50}$ | $AP_{75}$ | mAP | $AP_{50}$ | $AP_{75}$ | mAP | $AP_{50}$ | $AP_{75}$ |
| ZS | 26.9 | 47.9 | 25.9 | 24.4 | 40.1 | 26.2 | 17.8 | 29.4 | 18.8 | 18.8 | 31.0 | 17.9 | 13.3 | 22.0 | 13.4 | 35.2 | 64.7 | 34.6 |
| TPT | 27.3 | 48.5 | 26.1 | 24.9 | 41.3 | 26.8 | 18.1 | 29.9 | 19.1 | 18.8 | 31.1 | 18.0 | 13.4 | 22.2 | 13.5 | 35.8 | 65.1 | 34.7 |
| VPT | 26.9 | 49.1 | 25.1 | 25.0 | 41.4 | 26.9 | 18.3 | 30.9 | 19.3 | 18.9 | 31.2 | 18.0 | 13.5 | 22.3 | 13.2 | 35.8 | 65.8 | 34.9 |
| DPE | 27.2 | 48.9 | 26.3 | 24.9 | 41.5 | 27.1 | 18.9 | 31.7 | 19.8 | 19.0 | 31.3 | 18.0 | 13.5 | 22.3 | 13.3 | 35.9 | 66.4 | 35.1 |
| Adapter | 28.3 | 51.5 | 26.7 | 26.9 | 44.1 | 27.8 | 20.8 | 34.7 | 21.7 | 19.1 | 31.3 | 18.3 | 13.7 | 21.7 | 13.1 | 35.8 | 66.4 | 35.1 |
| **VLOD-TTA** | **29.6** | **53.1** | **28.7** | **28.1** | **45.4** | **29.9** | **21.4** | **36.1** | **22.1** | **19.4** | **31.8** | **18.6** | **14.6** | **24.3** | **14.8** | **36.4** | **67.4** | **35.6** |
| | Grounding DINO | | | | | | | | | | | | | | | | | |
| | Watercolor | | | ClipArt | | | Comic | | | Cityscapes | | | BDD100K | | | ExDark | | |
| Method | mAP | $AP_{50}$ | $AP_{75}$ | mAP | $AP_{50}$ | $AP_{75}$ | mAP | $AP_{50}$ | $AP_{75}$ | mAP | $AP_{50}$ | $AP_{75}$ | mAP | $AP_{50}$ | $AP_{75}$ | mAP | $AP_{50}$ | $AP_{75}$ |
| ZS | 37.4 | 62.9 | 37.6 | 38.4 | 58.8 | 41.8 | 31.2 | 52.9 | 31.5 | 24.1 | 38.2 | 24.5 | 16.6 | 28.3 | 16.2 | 35.4 | 66.2 | 34.5 |
| TPT | 37.4 | 63.1 | 37.6 | 38.6 | 59.1 | 42.3 | 31.5 | 53.6 | 31.8 | 24.6 | 38.6 | 24.8 | 16.6 | 28.4 | 16.2 | 35.6 | 66.5 | 34.9 |
| VPT | 37.2 | 63.0 | 37.8 | 38.6 | 59.0 | 41.9 | 31.1 | 52.6 | 31.2 | 24.0 | 38.2 | 24.3 | 16.7 | 28.5 | 16.2 | 35.1 | 66.4 | 34.4 |
| DPE | 37.6 | 63.2 | 38.1 | 38.2 | 59.3 | 42.0 | 31.8 | 53.3 | 31.7 | 24.4 | 38.3 | 24.4 | 16.7 | 28.6 | 16.3 | 35.2 | 66.6 | 34.3 |
| Adapter | 38.4 | 63.6 | 39.0 | 38.6 | 58.9 | 41.8 | 31.7 | 54.1 | 32.0 | 24.6 | 39.1 | 24.6 | 16.8 | 28.7 | 16.6 | 35.7 | 66.8 | 34.5 |
| **VLOD-TTA** | **38.9** | **64.7** | **39.5** | **41.2** | **62.1** | **43.3** | **34.2** | **57.8** | **35.3** | **25.8** | **40.8** | **25.9** | **18.1** | **31.1** | **18.5** | **37.3** | **68.9** | **36.8** |

Table 1: **Detection performance of TTA methods on benchmark datasets.** mAP, $AP_{50}$, and $AP_{75}$ for both **YOLO-World** and **Grounding DINO** ODs on six benchmark datasets – Watercolor, ClipArt, Comic, Cityscapes, BDD100K, and ExDark. Best results are highlighted in bold.

## 4.2 Main Results

**Performance analysis under texture and style shifts (*Watercolor, ClipArt, Comic*).** Results in Tab. 1 show that in YW, VPT is slightly more effective than TPT, with an average gain of +0.6 $AP_{50}$. In GD, TPT is more effective than VPT by +0.4 $AP_{50}$ on average. This pattern is consistent with our ablations (Sec. 4.3): YW benefits more from adapting the visual backbone, whereas GD benefits more from adapting the textual side, likely due to differences in text–vision fusion (single-stage in YW vs. multi-stage in GD). For both YW and GD, DPE provides slight improvements over prompt tuning, with +0.2 over VPT and +0.8 over TPT in YW, and +0.4 over both in GD. The Adapter baseline adds +4.3 $AP_{50}$ over ZS in YW, averaged over the three stylized sets, and +0.7 in GD. VLOD-TTA consistently outperforms all baselines on these stylized domains. In YW, the improvements over ZS are +3.3 mAP, +5.8 $AP_{50}$, and +3.2 $AP_{75}$. In GD, they are +2.4 mAP, +3.3 $AP_{50}$, and +2.4 $AP_{75}$, showing the effectiveness of VLOD-TTA across style shifts.

**Performance analysis on autonomous driving under various conditions (*Cityscapes, BDD100K*).** Tab. 1 shows that adapting VLODs to driving scenes is more challenging than stylized domains, likely due to the large number of small objects with low overlap (see App. A.2 for detailed analysis). In YW, all baselines yield only marginal gains. On BDD100K, the Adapter baseline is lower than ZS in both $AP_{50}$ and $AP_{75}$, highlighting a limitation of standard entropy for small objects. Our method achieves the highest scores, exceeding all baselines. Across Cityscapes and BDD100K, VLOD-TTA improves over ZS on YW by an average of +1.0 mAP, +1.6 $AP_{50}$, and +1.1 $AP_{75}$. On GD, the average gains over ZS across Cityscapes and BDD100K are +1.6 mAP, +2.7 $AP_{50}$, and +1.9 $AP_{75}$. These results indicate effectiveness in driving scenarios, even though absolute gains are smaller than under texture and style shifts.

**Performance analysis under illumination shift (*ExDark*).** From Tab. 1, it can be observed that the results in the low-light setting follow a similar trend to those under style shifts. On YW, DPE and Adapter both reach $AP_{50}$ of 66.4 and are close on mAP and $AP_{75}$, indicating that using prior information helps in low light. On GD, Adapter is the strongest prior baseline, yielding +0.6 $AP_{50}$ and +0.3 mAP over ZS, while the other baselines change the metrics only slightly. Our method improves over ZS by +1.2 mAP, +2.7 $AP_{50}$, and +1.0 $AP_{75}$ on YW, and by +1.9 mAP, +2.7 $AP_{50}$, and +2.3 $AP_{75}$ on GD, exceeding all baselines. These results indicate effectiveness under illumination shift, improving both recall at 50 IoU and localization at 75 IoU.

**Performance analysis under common corruptions.** Tab. 2 reports $AP_{50}$ on PASCAL-C across 15 corruption types using YW. VLOD-TTA attains the best score on every corruption and the highest average (38.5 $AP_{50}$), improving over the strongest baseline (Adapter) by +1.5 $AP_{50}$ on average and over ZS (34.6) by +3.9. The Adapter baseline overfits on Motion, Zoom, and Brightness corruptions, highlighting limitations of standard entropy. Our gains are consistent across corruption families, with notable improvements on JPEG Compression (+8.2), Glass Blur (+7.0), Contrast (+5.8), Elastic

| Method | Noise | | | Blur | | | | Weather | | | Digital | | | | | Avg |
|---|---|---|---|---|---|---|---|---|---|---|---|---|---|---|---|---|
| | Gaussi. | Shot | Impulse | Defocus | Glass | Motion | Zoom | Snow | Frost | Fog | Bright. | Contrast | Elastic | Pixel | JPEG | |
| ZS | 11.9 | 11.4 | 11.2 | 49.6 | 15.5 | 34.1 | 27.6 | 35.7 | 50.0 | 71.8 | 74.3 | 44.4 | 50.8 | 10.1 | 20.7 | 34.6 |
| TPT | 12.6 | 11.8 | 11.4 | 50.2 | 15.7 | 34.1 | 28.0 | 36.2 | 50.7 | 71.9 | 74.1 | 46.7 | 51.9 | 10.7 | 21.6 | 35.2 |
| VPT | 12.7 | 12.1 | 11.5 | 50.2 | 15.9 | 33.9 | 27.7 | 36.7 | 50.5 | 72.0 | 74.0 | 46.7 | 52.0 | 10.9 | 21.9 | 35.2 |
| DPE | 13.1 | 12.4 | 11.8 | 50.5 | 16.2 | 34.7 | 28.1 | 37.1 | 50.5 | 72.4 | 75.0 | 48.2 | 52.3 | 11.5 | 22.9 | 35.8 |
| Adapter | 14.7 | 15.2 | 13.2 | 51.5 | 18.8 | 33.0 | 26.4 | 39.7 | 52.4 | 71.8 | 73.0 | 47.8 | 55.4 | 14.8 | 28.7 | 37.0 |
| **VLOD-TTA** | **15.3** | **15.9** | **14.7** | **53.1** | **22.5** | **35.3** | **27.9** | **40.5** | **53.5** | **73.4** | **74.9** | **50.2** | **55.9** | **15.2** | **28.9** | **38.5** |

Table 2: **Detection performance of TTA methods on PASCAL-C.** $AP_{50}$ is reported for the **YOLO-World** detector on 15 different data corruptions. Best results are highlighted in bold.

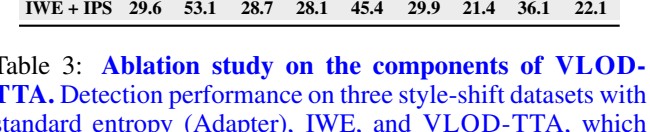

| Method | Watercolor | | | ClipArt | | | Comic | | |
|---|---|---|---|---|---|---|---|---|---|
| | mAP | $AP_{50}$ | $AP_{75}$ | mAP | $AP_{50}$ | $AP_{75}$ | mAP | $AP_{50}$ | $AP_{75}$ |
| Zero-shot | 26.9 | 47.9 | 25.9 | 24.4 | 40.1 | 26.3 | 17.8 | 29.4 | 18.8 |
| Adapter | 28.3 | 51.5 | 26.7 | 26.9 | 44.1 | 27.8 | 20.8 | 34.7 | 21.7 |
| IWE | 29.3 | 52.6 | 28.6 | 27.5 | 44.7 | 29.0 | 21.2 | 35.6 | 22.0 |
| **IWE + IPS** | **29.6** | **53.1** | **28.7** | **28.1** | **45.4** | **29.9** | **21.4** | **36.1** | **22.1** |

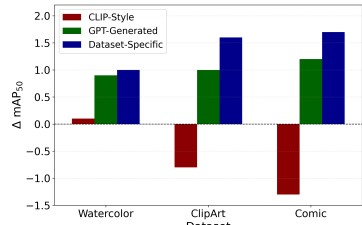

Table 3: **Ablation study on the components of VLOD-TTA.** Detection performance on three style-shift datasets with standard entropy (Adapter), IWE, and VLOD-TTA, which combines IWE with IPS.

Figure 4: **Prompt generation strategies.** $\Delta mAP_{50}$ over three style-shift datasets measured relative to ZS.

Transform (+5.1), and Pixelate (+5.1). These results indicate that combining IoU-weighted entropy with image-conditioned prompt selection yields robust benefits under various corruptions.

### 4.3 ABLATION STUDIES

**Contribution of individual components.** We ablate the two components of VLOD-TTA on YW using Watercolor, ClipArt, and Comic datasets, as shown in Tab. 3. Adapter denotes standard entropy minimization where only lightweight adapters are updated. IWE replaces the standard entropy loss with IoU-weighted entropy while keeping the same adaptation setup. VLOD-TTA combines IWE with IPS. Relative to Adapter, IWE consistently improves both localization (mAP, $AP_{75}$) and classification ($AP_{50}$) on all three datasets. Adding IPS on top of IWE consistently yields further gains and achieves the best overall performance. The results demonstrates that each component contributes positively, and the combination achieves the best performance.

**Variation in performance with different prompt-generation strategies.** GPT-generated prompts were used without any dataset information. In this ablation, we compare two alternatives, namely dataset-specific GPT prompts and CLIP-style prompts (see App. A.8). The improvement over ZS on Watercolor, ClipArt, and Comic is summarized in Fig. 4. CLIP-style prompts yield a small gain on Watercolor while reducing accuracy on ClipArt ($-0.8$) and Comic ($-1.3$). We hypothesize this reflects a pretraining bias toward label-only prompts. Our dataset-agnostic GPT prompts improve over ZS on all three datasets, with gains of $+0.9$ on Watercolor, $+1.0$ on ClipArt, and $+1.2$ on Comic. Dataset-specific prompts provide the largest gains, with $+1.0$ on Watercolor, $+1.6$ on ClipArt, and $+1.7$ on Comic, indicating that data-specific cues can further help adaptation of VLODs. Because dataset-specific knowledge is not ideal for a realistic TTA, *we did not use dataset-specific prompts in our experiments.*

**Adapters versus batch-normalization parameters.** In our main experiments, we optimize adapter parameters. To show that VLOD-TTA can also be used with other parameter subsets, we compare optimizing adapters and batch-normalization layers under both the standard entropy loss and VLOD-TTA in Tab. 4. With the standard entropy loss, both adapters and batch norm achieve similar gains over ZS on Watercolor, ClipArt, and Comic. Using VLOD-TTA further improves both parameter choices, consistently outperforming the standard entropy loss. Batch norm with VLOD-TTA almost matches the adapter baseline, showing that batch norm parameters can be used instead of adapters. However, batch norm requires dataset-specific learning-rate tuning to reach its best performance (for example $1e{-}2$ on Watercolor and $3e{-}2$ on ClipArt), whereas adapters work well with a single learning rate of $5e{-}3$ across datasets. In the TTA setting, the target domain is unknown

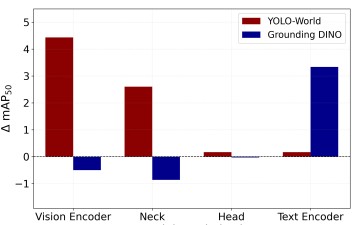

| Method | Watercolor | | | ClipArt | | | Comic | | |
|---|---|---|---|---|---|---|---|---|---|
| | mAP | $AP_{50}$ | $AP_{75}$ | mAP | $AP_{50}$ | $AP_{75}$ | mAP | $AP_{50}$ | $AP_{75}$ |
| Zero-shot | 26.9 | 47.9 | 25.9 | 24.4 | 40.1 | 26.3 | 17.8 | 29.4 | 18.8 |
| B.Norm | 28.4 | 51.3 | 26.5 | 26.7 | 44.1 | 28.1 | 20.6 | 34.5 | 21.7 |
| Adapters | 28.3 | 51.5 | 26.7 | 26.9 | 44.1 | 27.8 | 20.8 | 34.7 | 21.7 |
| **B.Norm VLOD-TTA** | 29.4 | 52.9 | **28.8** | 28.3 | 45.3 | **30.0** | 21.3 | **36.1** | 22.0 |
| **VLOD-TTA** | **29.6** | **53.1** | 28.7 | 28.1 | **45.4** | 29.9 | **21.4** | **36.1** | **22.1** |

Table 4: **Adapters vs batch norm as adaptation parameters.** Detection performance on three style-shift datasets.

Figure 5: **Adapters in different detector modules.** Mean $\Delta mAP_{50}$ averaged over three style-shift datasets, relative to ZS.

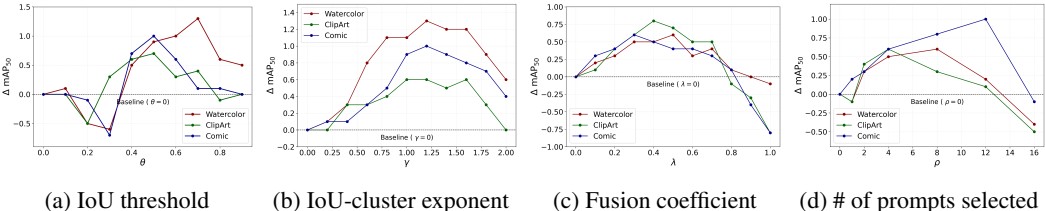

| (a) IoU threshold | (b) IoU-cluster exponent | (c) Fusion coefficient | (d) # of prompts selected |
|---|---|---|---|

Figure 6: **Variation in performance with hyperparameters on three style-shift datasets.** The IoU threshold ($\theta$) and IoU-cluster exponent ($\gamma$) influence the IWE, while the fusion coefficient ($\lambda$) and selection fraction ($\rho$) are IPS hyperparameters.

at test-time, so tuning the learning rate per-dataset is impractical for real-time object detection. For this reason, we choose to update adapter parameters in our main experiments.

**Effect of adapter placement across modules.** In Fig. 5, we insert adapters into one module at a time and report the change in $AP_{50}$ relative to ZS, averaged over Watercolor, ClipArt, and Comic. On YW, adapting the vision encoder yields the largest gain ($+4.4$ $AP_{50}$). The neck ranks second ($+2.6$), and the head and text encoder give negligible gains. On GD, adapting the text encoder yields the best gain ($+3.3$). The head changes little, and adapting the vision encoder or the neck slightly degrades performance. These trends indicate that the most effective use of adaptation capacity is to update the vision encoder on YW and the text encoder on GD, which is the configuration adopted in our main experiments.

### 4.4 HYPERPARAMETER SENSITIVITY ANALYSIS

We conduct sensitivity analyses on four crucial hyperparameters that influence the performance of VLOD-TTA. The results on three style-shift datasets are shown in Fig. 6.

**Effect of $\theta$ for graph construction.** The IoU threshold $\theta$ determines how proposals are clustered within a class. As shown in Fig. 6(a), as $\theta \to 0$, the proposals of a class coalesce into a single cluster, so the objective behaves similarly to standard entropy. Interestingly, performance drops when $\theta$ is around 0.2–0.3, likely because meaningful clusters are not formed at these thresholds. As $\theta \to 1$, proposals are rarely grouped, which diminishes the advantage of our method. Empirically, Watercolor (with slightly larger objects) attains its best performance around $\theta \approx 0.7$, and ClipArt and Comic (with smaller objects on average) peak near $\theta \approx 0.5$. Overall, while the optimal value shows mild dataset dependence, performance is generally stable for $0.5 \le \theta \le 0.7$.

**Effect of $\gamma$ for graph construction.** The exponent $\gamma$ controls how strongly component size influences the IoU-weighted entropy. When $\gamma=0$, all $w_i$ are equal and the objective reduces to standard entropy. As shown in Fig. 6(b), performance improves from $\gamma=0$ and peaks around $\gamma \approx 1.0$–1.2 across datasets. For very large $\gamma$, performance drops because large clusters can suppress small but correct objects. Overall, $\gamma \in [0.6, 1.6]$ is stable across datasets, with $\gamma \approx 1.0$–1.2 performing best in our experiments.

**Effect of $\lambda$ in prompt-selection.** The fusion coefficient $\lambda$ balances the selected-prompt score $\tilde{z}_{i,k}$ and the original detector score $s_{i,k}$ in Eq. (5). As shown in Fig. 6(c), performance rises from

| (a) GT | (b) ZS | (c) Adapter | (d) VLOD-TTA |

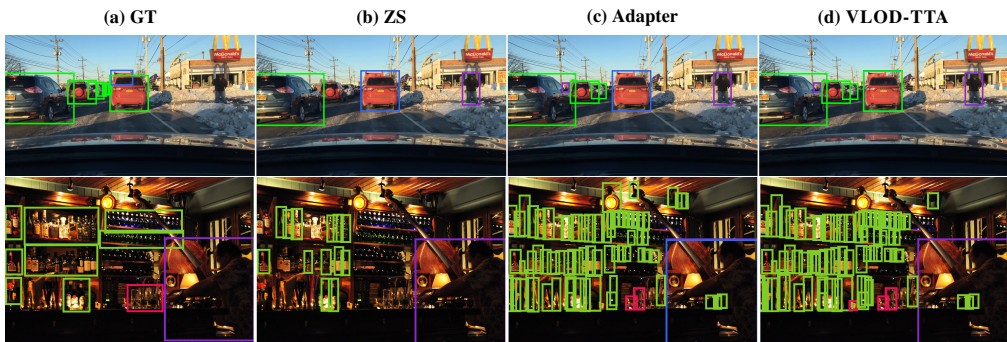

Figure 7: **YOLO-World detections across different approaches**: Each column corresponds to a different method: (a) GT (Ground Truth), (b) ZS (Zero-Shot), (c) Adapter, and (d) VLOD-TTA. Each color represents a different object category.

$\lambda = 0$ and peaks between $\lambda \approx 0.3$ and $0.5$ depending on the dataset. After that, performance decreases steadily and drops sharply as $\lambda \to 1$. This decline is due to early visual–text fusion in VLODs, which makes the region features partly dependent on the text embeddings, so relying solely on selected prompts discards useful information carried by the original detector prompts. In GD, where fusion occurs at multiple stages, the original detector score is even more important, and the best performance is obtained at $\lambda = 0.1$.

**Effect of $\rho$ in prompt-selection.** For each class, we keep the top-$\rho$ fraction of prompts by their similarity scores. As Fig. 6(d) shows, increasing $\rho$ from 0 initially improves performance by incorporating more informative templates, after which the gains saturate and ultimately decline as weaker templates are included. In general, very small $\rho$ underuses the prompt pool, very large $\rho$ adds noise, and $\rho \approx 0.25$–$0.5$ works best.

### 4.5 QUALITATIVE ANALYSIS

Fig. 7 compares ZS, the Adapter baseline, and our method. The Adapter baseline makes several incorrect or inconsistent predictions (e.g., the person in the middle row), highlighting the confirmation bias of standard entropy. By exploiting the structure of dense proposals, our method produces more accurate detections with fewer false positives. Interestingly, VLODs can correctly detect objects missing from the GT (e.g., the person in the top row) and refine loosely annotated boxes, which are shown in the figure but are counted as errors in quantitative evaluation.

## 5 CONCLUSION

TTA for VLMs has been studied extensively for classification, yet it remains largely unexplored for VLODs. In this paper, we close this gap by introducing VLOD-TTA, the first TTA framework for VLODs. Our approach combines IoU-weighted entropy with image-conditioned prompt selection to optimize lightweight adapter parameters. We show the robustness of VLOD-TTA by benchmarking across style shifts, driving scenes, low-light conditions, and common corruptions on two popular VLODs, YOLO-World and Grounding DINO. VLOD-TTA significantly outperforms baselines without additional training or annotation while preserving ZS capability.

Despite strong and consistent gains, IoU-weighted entropy can underperform in scenes dominated by many tiny, low-overlap objects (e.g., Cityscapes). Future work will explore efficient adaptation of alternative VLM-TTA objectives to VLODs.

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

# A APPENDIX

This appendix provides complementary theory, implementation specifics, and extended empirical results. We begin with a note on **Cosine–Euclidean equivalence** (App. A.1), then detail **adapter placement** (App. A.4) and analyze the IoU graph with **top-$M$ proposals** (App. A.5). We report **computational complexity** (App. A.6), **compare prompt averaging to image-conditioned prompt selection** (App. A.7), and list examples of **prompt sets** (App. A.8). Subsequent sections present ablations on **batch size** (App. A.9) and **augmentation** (App. A.10), robustness across **detector backbones** (App. A.11), and the **effect of pre-adaptation fine-tuning** (App. A.12), followed by comprehensive **COCO-C** (App. A.14) and **PASCAL-C** (App. A.15) results and qualitative analyses—including **standard entropy vs. IoU-weighted entropy** (App. A.16)—and additional **detection visualizations** (App. A.17).

## A.1 COSINE–EUCLIDEAN EQUIVALENCE

Let $\hat{\mathbf{v}}_i, \hat{\mathbf{e}}_{k,t} \in \mathbb{R}^d$ be $\ell_2$-normalized region features and prompt embeddings, i.e., $\|\hat{\mathbf{v}}_i\|_2 = \|\hat{\mathbf{e}}_{k,t}\|_2 = 1$. Define the per-proposal cosine similarity $z_{i,k,t} = \hat{\mathbf{v}}_i^\top \hat{\mathbf{e}}_{k,t} \in [-1, 1]$ and its image-level average $r_{k,t} = \frac{1}{N} \sum_{i=1}^N z_{i,k,t}$.

**Proposition.** The mean squared Euclidean distance between the region features and prompt $t$ satisfies the following equation:

$$\frac{1}{N} \sum_{i=1}^N \|\hat{\mathbf{v}}_i - \hat{\mathbf{e}}_{k,t}\|_2^2 = 2 - 2\, r_{k,t}.$$

Consequently, maximizing $r_{k,t}$ is equivalent to minimizing the mean squared Euclidean distance.

*Proof.* Since $\|\hat{\mathbf{v}}_i\|_2 = \|\hat{\mathbf{e}}_{k,t}\|_2 = 1$,

$$\|\hat{\mathbf{v}}_i - \hat{\mathbf{e}}_{k,t}\|_2^2 = \|\hat{\mathbf{v}}_i\|_2^2 + \|\hat{\mathbf{e}}_{k,t}\|_2^2 - 2\, \hat{\mathbf{v}}_i^\top \hat{\mathbf{e}}_{k,t} = 2 - 2\, z_{i,k,t}.$$

Averaging over $i$ gives

$$\frac{1}{N} \sum_{i=1}^N \|\hat{\mathbf{v}}_i - \hat{\mathbf{e}}_{k,t}\|_2^2 = \frac{1}{N} \sum_{i=1}^N (2 - 2\, z_{i,k,t}) = 2 - 2\, r_{k,t}.$$

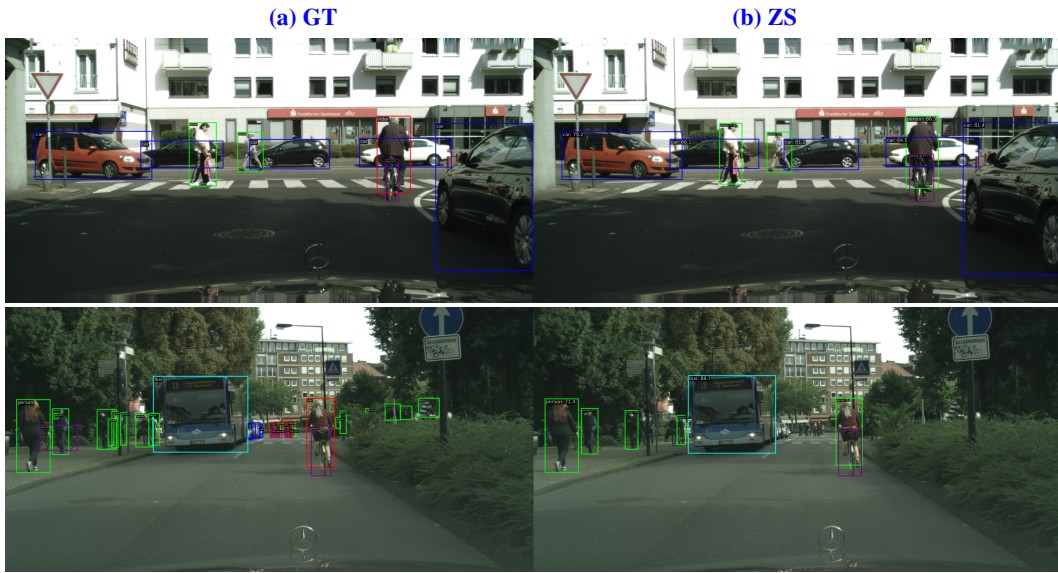

Figure 8: **Cityscapes detection with ground-truth (GT) and zero-shot (ZS). The ZS model detects rider (red) as person (green)**

| Cityscapes | Person | Rider |
|---|---|---|
| ZS | 17.8 | 6.3 |
| **VLOD-TTA** | 16.5 | 1.9 |
| ZS (merged rider) | 23.2 | – |
| **VLOD-TTA (merged rider)** | **24.4** | – |

Table 5: **Class-wise AP on Cityscapes.** AP for person and rider before and after merging rider into person.

| Cityscapes | mAP | mAP$_{50}$ | mAP$_{75}$ |
|---|---|---|---|
| ZS | 18.8 | 31.0 | 17.9 |
| **VLOD-TTA** | 19.4 | 31.8 | 18.6 |
| ZS (merged rider) | 21.3 | 34.7 | 20.5 |
| **VLOD-TTA (merged rider)** | **22.5** | **35.9** | **21.1** |

Table 6: **Effect of merging rider into person on Cityscapes.** Detection performance with the original 8-class labels and with rider merged into person.

## A.2 CITYSCAPES FAILURE-CASE ANALYSIS.

Our method underperforms on Cityscapes dataset, so we perform a detailed ablation to identify the main causes. We find two factors: label overlap between rider and person, and a large proportion of small objects.

**1. Overlap between rider and person.** Cityscapes has eight classes, including rider and person. In practice, riders are visually very similar to persons, and the ZS model is biased toward the person label. As a result, a rider is often localized correctly but predicted as person as shown in Fig. 8. Our IoU-weighted entropy (IWE) further reinforces this behavior by sharpening high-IoU clusters, which pushes ambiguous rider–person cases toward the person label. This is still counted as an error under the dataset labels, even though the localization is correct. This effect is visible in Tab. 5, where we report class-wise AP for person and rider separately. Compared to the ZS model, VLOD-TTA reduces AP for both classes.

**Ablation -** We merge the rider class into person to create a 7-class annotation set. As shown in Tab. 6, the ZS baseline improves after mapping rider to person, and the class-wise AP in Tab. 5 confirms that this corresponds to a higher AP for the merged person class. VLOD-TTA further improves in this setting because IWE no longer has to distribute probability between rider and person, so entropy minimization concentrates on unified person clusters and yields larger gains.

**2. Small objects and input resolution.** Cityscapes contains many small objects and the original resolution is 2048×1024. YOLO-World resizes inputs to 640×640 by default, which further shrinks distant cars and pedestrians and leads to missed detections. At this resolution, many small objects do not generate stable overlapping proposals, so IWE underperforms. Qualitative examples are shown in Fig. 9.

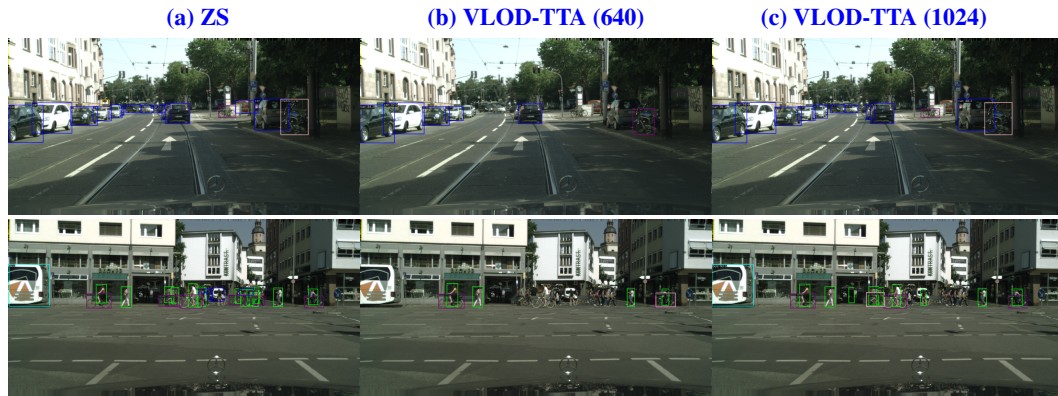

Figure 9: **Cityscapes detections with zero-shot (ZS) and VLOD-TTA at image scales 640×640 and 1024×1024.** VLOD-TTA (1024) detects more small objects than VLOD-TTA (640).

| Cityscapes | mAP | mAP$_{50}$ | mAP$_{75}$ |
|---|---|---|---|
| ZS (640) | 18.8 | 31.0 | 17.9 |
| **VLOD-TTA (640)** | 19.4 | 31.8 | 18.6 |
| ZS (1024) | 28.5 | 43.9 | 28.3 |
| **VLOD-TTA (1024)** | **31.2** | **46.6** | **30.5** |

Table 7: **Effect of input resolution on Cityscapes.** Detection performance at 640×640 and 1024×1024 input resolutions.

**Ablation -** We evaluate the model at 1024×1024 input resolution. As shown in Tab. 7, the zero-shot baseline increases substantially with higher resolution. VLOD-TTA then improves further, with a modest gain at 640 (+0.6 mAP) and a much larger gain at 1024 (+2.7 mAP). At higher resolution, more small objects produce stable overlapping proposals that IWE can weight and sharpen during adaptation, confirming that small objects are a key factor behind the original underperformance, as illustrated in Fig. 9.

**Conclusion.** The Cityscapes underperformance stems from label overlap between rider and person and from the prevalence of small objects at low input resolution. Once we merge rider into person and increase the resolution, VLOD-TTA yields substantially larger gains on Cityscapes.

## A.3 IoU-WEIGHTING WITH PSEUDO LABELING

To further demonstrate the applicability of our IoU-weighting beyond entropy minimization, we integrate it into a standard pseudo label based TTA scheme for ODs. Pseudo labeling is a widely used TTA strategy with a mean teacher setup, where the teacher generates pseudo labels that supervise student updates. To obtain IoU-weighted pseudo labels, we cluster overlapping teacher boxes by IoU and weight each pseudo label by its normalized cluster size. **??** compares standard pseudo labeling with its IoU weighted variant on the Watercolor, ClipArt, and Comic datasets. IoU weighted pseudo labeling consistently improves mAP, mAP$_{50}$, and mAP$_{75}$ across all three datasets. This shows that IoU-based weighting can be plugged into existing pseudo label losses rather than being restricted to our entropy minimization objective.

## A.4 ADAPTER PLACEMENT AND CONFIGURATION

For YOLO-World, adapters are inserted after every convolution in the backbone and neck (Chen et al., 2024). Concretely, for each convolutional block with feature map $x \in \mathbb{R}^{C \times H \times W}$, we append a lightweight residual path composed of a 1×1 down-projection to $C/r$ channels, a depthwise $k \times k$ convolution, and a 1×1 up-projection back to $C$, with the result added to $x$. The final 1×1 is zero-initialized, so the adapter path is an identity at initialization. During adaptation, all pre-trained detector weights are frozen, and only adapter parameters are optimized. Unless otherwise stated, we use $r=4$ and $k=3$. These adapters are attached after every ConvModule throughout the CSPDarknet backbone and the neck.

|        | Watercolor | | | ClipArt | | | Comic | | |
| --- | --- | --- | --- | --- | --- | --- | --- | --- | --- |
| Method | mAP | $AP_{50}$ | $AP_{75}$ | mAP | $AP_{50}$ | $AP_{75}$ | mAP | $AP_{50}$ | $AP_{75}$ |
| Zero-shot | 26.9 | 47.9 | 25.9 | 24.4 | 40.1 | 26.3 | 17.8 | 29.4 | 18.8 |
| Pseudo-label | 28.3 | 50.1 | 27.0 | 25.9 | 42.3 | 28.1 | 19.2 | 31.9 | 20.1 |
| **IoU-weighted Pseudo-label (IWPL)** | **29.1** | **51.6** | **27.9** | **27.3** | **43.7** | **29.0** | **20.5** | **33.2** | **21.2** |

Table 8: **IoU-weighted pseudo labeling.** Integrating IoU-weighting into a mean-teacher pseudo-label TTA scheme improves detection AP on three style-shift datasets.

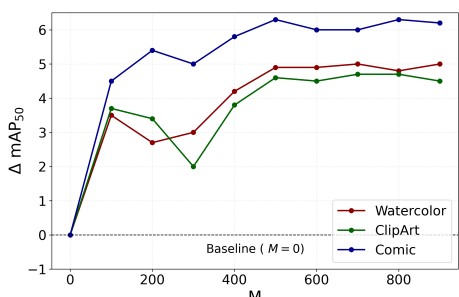

Figure 10: **Effect of top-$M$ proposals on three style-shift datasets.** IWE uses the top-$M$ proposals to construct the IoU graph.

For Grounding DINO, adapters are inserted after the output sublayer of every Transformer block in the BERT encoder (Houlsby et al., 2019). Each adapter is a two-layer bottleneck MLP with GELU, added residually to the layer output, and the up-projection is zero-initialized to preserve the pre-trained function at the start of adaptation. We use a bottleneck reduction ratio $r$, meaning that for hidden size $d$ the adapter hidden width is $d/r$ (we use $r=16$ unless otherwise stated). We disable dropout in the language backbone, freeze all BERT weights, and update only adapter parameters. Text features are computed exactly as in the baseline (average of the last $K$ hidden layers, with $K=1$ in our experiments).

Parameter overhead is small: per convolution with $C$ output channels, the vision adapter adds $\frac{2C^2}{r} + \frac{C}{r}k^2$ parameters, and per Transformer layer with hidden size $d$ the language adapter adds $\frac{2d^2}{r}$ parameters. Zero-initialization ensures no degradation at the start of adaptation.

### A.5 IoU Graph: Effect of Top-$M$ Proposals

Our method selects the top-$M$ proposals to build the IoU graph. This is done to speed up the construction of the IoU graph and to suppress extremely noisy boxes. As shown in Fig. 10, when $M$ is too small, improvements over ZS are limited because the resulting graph fails to capture the structure of the proposals. Performance peaks around $M = 600$, after which further increases yield no noticeable improvement.

### A.6 Complexity Analysis

Our method improves the robustness of VLODs at inference time, but it also introduces overhead. It increases per-image latency due to a backward pass and adds a lightweight adapter. In this section, we report the added parameters for YOLO-World and the frames per second (FPS). In Tab. 9, we compare FPS and trainable parameters against baselines and include full fine-tuning (FFT) for reference. Our method runs slower than the ZS detector due to the backward pass, yet it remains slightly faster than most baselines. The Adapter baseline is marginally faster than ours because our approach builds an IoU graph that adds computation. TPT and VPT are slower than our method because they require backpropagation through the entire text encoder and detector, respectively. DPE is much slower than our method, as it requires an iteration over proposals to update the cache memory. In terms of parameter budget, the adapter adds only a small number of parameters compared to FFT.

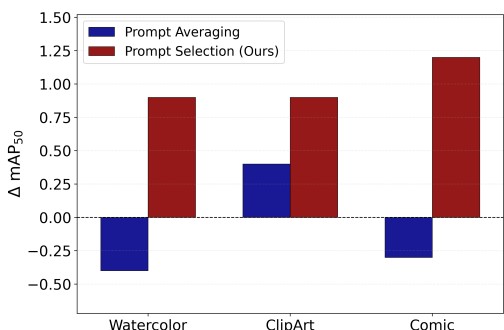

Figure 11: **Comparison of Prompt Averaging and Prompt Selection.** $\Delta$mAP$_{50}$ over three style-shift datasets measured relative to ZS.

| Method | FPS | Train Params (M) |
|---|---|---|
| ZS | 89 | 0.00 |
| FFT | 89 | 76.81 |
| TPT | 9 | 1.12 |
| VPT | 18 | 3.93 |
| DPE | 15 | 0.31 |
| Adapter | 22 | 1.52 |
| Ours | 20 | 1.61 |

Table 9: **Comparison of throughput and training parameters on YOLO-World.** FPS (frames per second; higher is better). Trainable parameters are in millions.

Overall, our approach offers a favorable robustness–cost trade-off, delivering consistent gains with modest latency and a small parameter budget.

### A.7 PROMPT AVERAGING VS. PROMPT SELECTION

In Fig. 11, we compare two ways to leverage language supervision in YOLO-World: (i) *Prompt Averaging*, which averages multiple templates per class into a single prototype (Radford et al., 2021), and (ii) *Prompt Selection* (ours), which selects the most informative prompts for the image. Averaging underperforms on two datasets and offers only a modest gain on ClipArt compared to the ZS baseline: AP$_{50}$ drops on Watercolor ($-0.35$) and Comic ($-0.30$) and rises slightly on ClipArt. Image-conditioned prompt selection improves all three datasets, with an average AP$_{50}$ gain of $+1.0$ across the three datasets. These trends indicate that averaging templates yields little benefit for VLODs, whereas per-image selection provides consistent, cross-dataset improvements.

### A.8 PROMPT EXAMPLES

Our prompt selection module chooses relevant prompts from a pool of candidates. In CLIP (Radford et al., 2021), prompts often follow generic templates such as "a photo of <class>" or "an origami of <class>". For VLODs, we observe that these templates are not effective. Using synonyms or verb-centric phrases gives slightly better results, so we use a GPT model to generate such candidates. The prompts used in our main experiments do not include any dataset-specific cues, but in one ablation, we use dataset-specific prompts. To generate these, we provide a set of training images to the GPT model and specify the dataset name. Tab. 10 shows examples for three classes on the ClipArt dataset across the three strategies.

### A.9 EFFECT OF BATCH SIZE

We use a batch size of 1 in our main experiments because it reflects a practical TTA setting. In Fig. 12a, we ablate over batch size. Across all three datasets, we observe the same trend. Performance rises slightly as the batch size increases to about 4–8, which indicates that our approach is not restricted to a batch size of 1. Beyond a batch size of 16, performance drops slightly. A likely reason is that the growing number of proposals makes entropy minimization less selective, so the optimization struggles to focus on the correct classes.

### A.10 EFFECT OF AUGMENTATIONS

In this section, we study how adding augmentations to our method affects performance. In these experiments, we used only scale augmentations, which we found most effective in preliminary tests. Augmentations are added in the order that performed best in preliminary tests. Results in Fig. 12b across the three style-shift datasets show that adding a single augmentation improves AP$_{50}$ by $+6.0$, $+6.6$, and $+5.1$ on Watercolor, ClipArt, and Comic. This indicates that our approach benefits from

| Prompt Strategy | Classes | | |
| --- | --- | --- | --- |
| | Aeroplane | Bicycle | Bird |
| CLIP-Style | "aeroplane", "a photo of an aeroplane", "a photograph of an aeroplane", "an image of an aeroplane", "a picture of an aeroplane", "a close-up photo of an aeroplane", "a cropped photo of an aeroplane", "a low-angle photo of an aeroplane", "a high-angle photo of an aeroplane", "a side view of an aeroplane", "a front view of an aeroplane", "a rear view of an aeroplane", "a black and white photo of an aeroplane", "a blurry photo of an aeroplane", "a bright photo of an aeroplane", "a dark photo of an aeroplane" | "bicycle", "a photo of a bicycle", "a photograph of a bicycle", "an image of a bicycle", "a picture of a bicycle", "a close-up photo of a bicycle", "a cropped photo of a bicycle", "a low-angle photo of a bicycle", "a high-angle photo of a bicycle", "a side view of a bicycle", "a front view of a bicycle", "a rear view of a bicycle", "a black and white photo of a bicycle", "a blurry photo of a bicycle", "a bright photo of a bicycle", "a dark photo of a bicycle" | "bird", "a photo of a bird", "a photograph of a bird", "an image of a bird", "a picture of a bird", "a close-up photo of a bird", "a cropped photo of a bird", "a low-angle photo of a bird", "a high-angle photo of a bird", "a side view of a bird", "a front view of a bird", "a rear view of a bird", "a black and white photo of a bird", "a blurry photo of a bird", "a bright photo of a bird", "a dark photo of a bird" |
| GPT-Generated | "aeroplane", "an airplane", "a passenger jet", "a commercial airliner", "a propeller plane", "a small aircraft", "a jet aircraft", "an aircraft taking off", "an aircraft landing", "a plane in flight", "a plane on the runway", "a twin-engine plane", "a private jet", "a cargo plane", "a jetliner", "an air transport aircraft" | "bicycle", "a pedal bicycle", "a road bike", "a mountain bike", "a commuter bicycle", "a racing bike", "a city bicycle", "a bike with basket", "a kids bike", "a fixed-gear bike", "a folding bicycle", "an electric bicycle", "a touring bike", "a parked bicycle", "a BMX bike", "a two-wheeled cycle" | "bird", "a flying bird", "a small bird", "a songbird", "a seabird", "a waterfowl", "a raptor", "a perching bird", "a wading bird", "a wild bird", "a bird in flight", "a perched bird", "a migratory bird", "a backyard bird", "a shorebird", "an avian animal" |
| Dataset-specific | "aeroplane", "cartoon airplane", "vector airplane", "flat-color airplane", "outlined airplane", "clip-art airplane", "airplane icon", "airplane silhouette", "bold-outline airplane", "comic-style airplane", "line-art airplane", "solid-fill airplane", "two-tone airplane", "SVG-style airplane", "white-background airplane", "no-texture airplane" | "bicycle", "cartoon bicycle", "vector bicycle", "flat-color bicycle", "outlined bicycle", "clip-art bicycle", "bicycle icon", "bicycle silhouette", "bold-outline bicycle", "comic-style bicycle", "line-art bicycle", "solid-fill bicycle", "two-tone bicycle", "SVG-style bicycle", "white-background bicycle", "no-texture bicycle" | "bird", "cartoon bird", "vector bird", "flat-color bird", "outlined bird", "clip-art bird", "bird icon", "bird silhouette", "bold-outline bird", "comic-style bird", "line-art bird", "solid-fill bird", "two-tone bird", "SVG-style bird", "white-background bird", "no-texture bird" |

Table 10: **Examples of prompts by strategy.** Prompt examples for three classes on the ClipArt dataset.

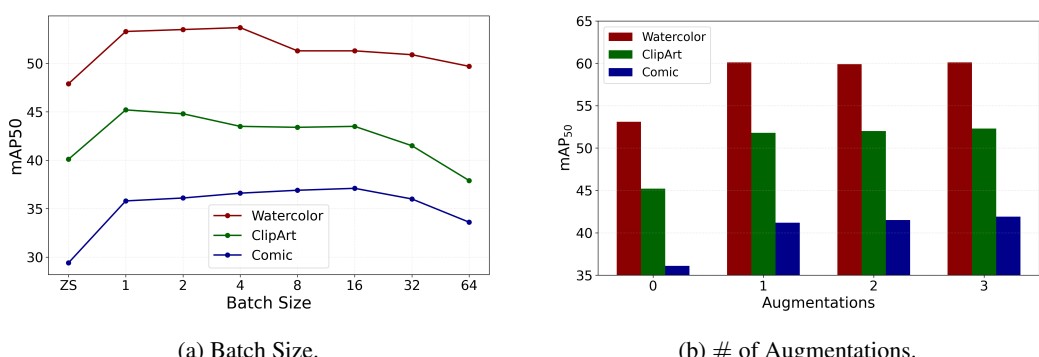

(a) Batch Size.                    (b) # of Augmentations.

Figure 12: **Effect of batch size and number of augmentations on performance.** We report $mAP_{50}$ across three style-shift datasets on YOLO-World.

modest augmentation. Adding more than one augmentation did not yield further gains in our setting, though tailoring the augmentation type and magnitude to each dataset may yield larger improvements.

## A.11 VARIATION IN PERFORMANCE WITH BACKBONE

In Tab. 11, we evaluate the effect of the detector backbone by applying our method to YOLO-World-Large (YW-L) and Grounding DINO-Big (GD-B). Across Watercolor, ClipArt, and Comic, both models show consistent improvements over the ZS baselines in mAP, $AP_{50}$, and $AP_{75}$. Although GD-B achieves higher absolute scores, the relative gains from our adaptation are similar for both backbones, indicating that our method is not tied to any specific architecture. Overall, the results demonstrate backbone-agnostic robustness from IoU-weighted entropy and image-conditioned prompt selection.

| YOLO-World-L | | | | | | | | |
| --- | --- | --- | --- | --- | --- | --- | --- | --- |
| **Watercolor** | | | **ClipArt** | | | **Comic** | | |
| **Method** mAP | AP$_{50}$ | AP$_{75}$ | mAP | AP$_{50}$ | AP$_{75}$ | mAP | AP$_{50}$ | AP$_{75}$ |
| ZS 32.8 | 55.3 | 33.0 | 31.1 | 50.6 | 32.6 | 23.3 | 37.9 | 23.6 |
| **Ours** 34.1 | 58.3 | 34.2 | 33.2 | 53.9 | 34.3 | 26.5 | 42.8 | 27.2 |
| **Grounding DINO-B** | | | | | | | | |
| **Watercolor** | | | **ClipArt** | | | **Comic** | | |
| **Method** mAP | AP$_{50}$ | AP$_{75}$ | mAP | AP$_{50}$ | AP$_{75}$ | mAP | AP$_{50}$ | AP$_{75}$ |
| ZS 42.6 | 70.5 | 44.6 | 53.0 | 77.9 | 58.8 | 38.7 | 64.7 | 39.2 |
| **Ours** 44.7 | 72.8 | 46.9 | 55.7 | 81.2 | 60.9 | 39.9 | 67.1 | 41.1 |

Table 11: **Detection performance with YOLO-World-L and Grounding DINO-B.** We report performance compared with ZS on three style-shift datasets.

## A.12 EFFECT OF FINE-TUNING VLODS BEFORE ADAPTATION

TTA methods for OD (Chen et al., 2023; Ruan & Tang, 2024) first fine-tune the detector on a source domain closer to the target domain before adaptation. For VLODs, this step is not required, since they already show strong ZS performance on most datasets. We study the effect of fine-tuning followed by adaptation and assess whether this step is beneficial for VLODs. We consider two cases: First, the source domain is PASCAL VOC (Everingham et al., 2010) and the target domains are Watercolor, Comic, and ClipArt (Inoue et al., 2018). This setting is popular in OD domain adaptation and provides a significant domain shift that challenges adaptation. Second, the source domain is COCO (Lin et al., 2014) and the target domain is COCO-C. This setting is slightly less challenging, since adaptation occurs within the same dataset; only the synthetic corruptions differ.

The results for the two settings are reported in Tabs. 12 and 13. We observe two patterns for the two settings. Fine-tuning on PASCAL VOC reduces the generalization ability of the model, and the performance drops on the three style-shift domains. Although our method improves over ZS in both cases by a similar margin, the absolute AP with fine-tuning is lower than without fine-tuning. In contrast, fine-tuning on COCO increases ZS on COCO-C. This is likely due to the small domain shift between the source and target and the large amount of training data. Our method also improves over ZS in this setting, which demonstrates its effectiveness. In summary, fine-tuning VLODs helps when training data are abundant and the target domain is close to the source, but it adds a training step and computational cost.

| No Fine-tune (PASCAL VOC AP$_{50}$ = 78.6) | | | | | | | | | | | |
| --- | --- | --- | --- | --- | --- | --- | --- | --- | --- | --- | --- |
| **Watercolor** | | | **ClipArt** | | | **Comic** | | | **Avg** | | |
| **Method** mAP | AP$_{50}$ | AP$_{75}$ | mAP | AP$_{50}$ | AP$_{75}$ | mAP | AP$_{50}$ | AP$_{75}$ | mAP | AP$_{50}$ | AP$_{75}$ |
| ZS 26.9 | 47.9 | 25.9 | 24.4 | 40.1 | 26.2 | 17.8 | 29.4 | 18.8 | 23.0 | 39.1 | 23.6 |
| **Ours** 29.6 | 53.1 | 28.7 | 28.1 | 45.2 | 29.9 | 21.4 | 36.1 | 22.1 | 26.4 | 44.8 | 26.9 |
| **Fine-tune (PASCAL VOC AP$_{50}$ = 82.3)** | | | | | | | | | | | |
| **Watercolor** | | | **ClipArt** | | | **Comic** | | | **Avg** | | |
| **Method** mAP | AP$_{50}$ | AP$_{75}$ | mAP | AP$_{50}$ | AP$_{75}$ | mAP | AP$_{50}$ | AP$_{75}$ | mAP | AP$_{50}$ | AP$_{75}$ |
| ZS 25.3 | 44.3 | 25.8 | 23.9 | 39.3 | 25.2 | 15.3 | 24.8 | 16.3 | 21.5 | 36.1 | 22.4 |
| **Ours** 27.8 | 49.8 | 27.8 | 25.6 | 44.2 | 28.6 | 19.1 | 31.1 | 19.2 | 24.2 | 41.7 | 25.2 |

Table 12: **Effect of fine-tuning before adaptation on YOLO-World.** For the No Fine-Tune, we use the YOLO-World pretrained model for adaptation, and for Fine-Tune, we first fine-tune the pretrained model on Pascal VOC before adaptation. We report our performance against ZS on three style-shift datasets. AP$_{50}$ on Pascal VOC for both settings is reported in the top row.

## A.13 EFFECTIVENESS OF VLOD-TTA ON SPECIALIZED DOMAINS.

To assess whether VLOD-TTA remains effective beyond generic benchmarks, we evaluate it on the Aquarium Object Detection dataset (Roboflow, 2020), which focuses on underwater animals such as fish, jellyfish, penguin, puffin, shark, starfish, and stingray. These categories are rare in standard

| | No Fine-tune (COCO $AP_{50}$ = 51.9) | | | | | | | | | | | | | | |
|---|---|---|---|---|---|---|---|---|---|---|---|---|---|---|---|
| | Noise | | | Blur | | | | Weather | | | Digital | | | | | Avg |
| Method | Gaussi. | Shot | Impulse | Defocus | Glass | Motion | Zoom | Snow | Frost | Fog | Bright. | Contrast | Elastic | Pixel | JPEG | Avg |
| ZS | 7.8 | 7.4 | 6.7 | 22.6 | 6.1 | 13.4 | 10.1 | 23.5 | 32.0 | 45.9 | 47.3 | 27.5 | 30.2 | 6.2 | 14.0 | 20.0 |
| **Ours** | **9.3** | **10.2** | **8.9** | **25.2** | **8.9** | **14.8** | **11.8** | **25.6** | **36.2** | **48.1** | **49.1** | **30.7** | **34.1** | **17.5** | **19.6** | **23.3** |

| | Fine-tune (COCO $AP_{50}$ = 57.8) | | | | | | | | | | | | | | |
|---|---|---|---|---|---|---|---|---|---|---|---|---|---|---|---|---|
| | Noise | | | Blur | | | | Weather | | | Digital | | | | | Avg |
| Method | Gaussi. | Shot | Impulse | Defocus | Glass | Motion | Zoom | Snow | Frost | Fog | Bright. | Contrast | Elastic | Pixel | JPEG | Avg |
| ZS | 13.7 | 13.3 | 12.6 | 26.2 | 9.6 | 18.0 | 11.9 | 27.9 | 37.2 | 52.1 | 52.9 | 30.3 | 34.5 | 12.7 | 19.4 | 24.8 |
| **Ours** | **15.5** | **14.6** | **14.5** | **27.8** | **12.6** | **19.6** | **12.9** | **29.2** | **38.8** | **53.5** | **53.8** | **33.2** | **36.7** | **19.3** | **24.1** | **27.1** |

Table 13: **Effect of fine-tuning before adaptation on YOLO-World.** For the No Fine-Tune, we use the YOLO-World pretrained model for adaptation, and for Fine-Tune, we first fine-tune the pretrained model on COCO before adaptation. We report our $mAP_{50}$ against ZS on 15 different corruptions. $AP_{50}$ on COCO for both settings is reported in the top row.

| Aquarium | mAP | $mAP_{50}$ | $mAP_{75}$ |
|---|---|---|---|
| Zero-shot | 11.9 | 20.3 | 11.6 |
| CLIP-Style Prompts | 11.6 | 19.7 | 11.2 |
| IPS | 12.4 | 21.5 | 12.1 |
| **VLOD-TTA** | **14.5** | **25.1** | **14.8** |

Table 14: **Effectiveness of VLOD-TTA on a specialized domain.** Detection performance on the Aquarium Object Detection dataset.

benchmarks and form a specialized domain that is challenging for prompt selection. As shown in Tab. 14, CLIP-Style Prompts degrades the ZS performance demonstrating it's ineffectiveness for ODs. In contrast IPS improves performance over the ZS baseline across all metrics, and combining IPS with IWE in VLOD-TTA yields further gain. This indicates that VLOD-TTA remains effective even in specialized domains.

## A.14 RESULTS ON COCO-C

We evaluate our approach on the COCO-C (Michaelis et al., 2019) benchmark across five corruption severities and fifteen corruption types. Full results are reported in Tabs. 15 to 17. Compared with PASCAL-C (Michaelis et al., 2019), COCO-C contains 80 categories, which makes test-time adaptation (TTA) more challenging. Overall, the zero-shot (ZS) YOLO-World baseline degrades consistently as severity increases for nearly all corruptions. For certain corruptions, e.g., Gaussian, Shot, Impulse noise, and the Pixelate transform, the baseline mAP can approach zero at high severity, underscoring the need for TTA in vision–language object detection (VLOD). TPT and VPT baselines also overfit on this dataset and sometimes perform worse than ZS, highlighting the difficulty of adapting COCO-C. The DPE baseline likewise struggles, with only marginal gains over ZS. In several cases, standard entropy minimization is the best-performing baseline and can marginally outperform our method on specific corruptions. Our method improves upon the ZS baseline in every setting—mAP, $mAP_{50}$, and $mAP_{75}$—across all corruption types and severities. The most noticeable gains are observed within the Digital Corruptions. Across severity levels, gains are most pronounced at severities 2–4. At severity 1, the improvements are smaller, as the test distribution remains close to the training distribution. At severity 5, performance is severely degraded for all methods, making most predictions unreliable, so it is challenging to improve. Nonetheless, our approach still yields consistent positive gains over ZS. These results demonstrate that the proposed TTA strategy substantially enhances robustness on COCO-C, particularly under moderate corruption, while still providing benefits under extreme distribution shifts.

## A.15 ADDITIONAL RESULTS ON PASCAL-C

Tabs. 18 and 19 report mAP and $AP_{75}$ for our method and baselines on PASCAL-C. Across all 15 corruptions, our approach outperforms the ZS baseline consistently, mirroring the trend observed at $AP_{50}$ for the same dataset. Gains are evident across the Noise, Blur, Weather, and Digital families,

| Sev. | Method | Noise | | | Blur | | | | Weather | | | Digital | | | | | Avg |
|---|---|---|---|---|---|---|---|---|---|---|---|---|---|---|---|---|---|
| | | Gaussi. | Shot | Impulse | Defocus | Glass | Motion | Zoom | Snow | Frost | Fog | Bright. | Contrast | Elastic | Pixel | JPEG | |
| 1 | ZS | 29.0 | 29.3 | 25.2 | 32.6 | 28.8 | 30.2 | 13.4 | 27.3 | 32.2 | 34.8 | 36.7 | 34.9 | 32.2 | 24.8 | 26.8 | 29.2 |
| | TPT | 27.2 | 27.5 | 23.3 | 30.8 | 28.8 | 29.8 | 12.5 | 26.7 | 31.2 | 33.9 | 36.0 | 34.0 | 31.5 | 24.7 | 26.6 | 28.3 |
| | VPT | 28.3 | 28.6 | 24.7 | 31.1 | 28.5 | 29.3 | 12.9 | 26.8 | 31.1 | 33.5 | 35.5 | 33.5 | 31.5 | 24.9 | 26.6 | 28.5 |
| | DPE | 28.8 | 29.0 | 25.1 | 32.8 | 29.3 | 30.5 | 13.7 | 27.5 | 32.3 | 34.7 | 36.5 | 35.1 | 32.4 | 25.0 | 27.2 | 29.3 |
| | Adapter | 28.7 | 29.1 | 25.8 | 32.3 | **29.5** | 30.6 | 13.7 | 28.0 | 32.4 | 34.8 | 36.2 | 34.7 | 32.8 | 28.4 | 28.7 | 29.7 |
| | **Our** | **29.6** | **30.1** | **26.1** | **33.8** | 29.4 | **31.1** | **14.5** | **28.6** | **33.1** | **36.1** | **38.2** | **35.7** | **33.0** | **29.7** | **29.5** | **30.6** |
| 2 | ZS | 22.9 | 22.8 | 18.8 | 29.3 | 21.7 | 23.1 | 8.3 | 19.6 | 26.6 | 33.9 | 35.9 | 33.2 | 28.8 | 17.1 | 20.4 | 24.2 |
| | TPT | 21.3 | 21.4 | 17.2 | 28.8 | 21.7 | 23.0 | 8.7 | 19.4 | 26.0 | 33.0 | 35.2 | 32.4 | 28.1 | 17.3 | 20.5 | 23.6 |
| | VPT | 22.8 | 22.8 | 18.9 | 27.8 | 21.8 | 22.4 | 8.0 | 19.5 | 25.7 | 32.5 | 34.7 | 31.9 | 28.0 | 17.7 | 20.6 | 23.7 |
| | DPE | 23.0 | 23.2 | 19.3 | 29.4 | 22.2 | 23.4 | 8.5 | 19.7 | 26.4 | 34.0 | 35.7 | 33.3 | 29.4 | 17.6 | 20.5 | 24.4 |
| | Adapter | 22.8 | 23.3 | 20.1 | 29.1 | 24.5 | 23.8 | 8.5 | 21.0 | 27.8 | 34.1 | 35.3 | 33.2 | 30.1 | 21.0 | 23.6 | 25.2 |
| | **Our** | **23.7** | **23.5** | **20.4** | **31.0** | **24.7** | **24.6** | **9.6** | **21.4** | **27.9** | **35.4** | **37.5** | **35.4** | **30.2** | **24.1** | **24.0** | **26.2** |
| 3 | ZS | 13.7 | 14.9 | 13.5 | 22.0 | 6.3 | 14.6 | 6.3 | 20.0 | 23.0 | 32.9 | 35.0 | 29.7 | 23.4 | 7.7 | 16.8 | 18.7 |
| | TPT | 12.6 | 13.6 | 13.3 | 21.8 | 5.5 | 14.6 | 6.5 | 19.6 | 23.3 | 32.0 | 34.3 | 29.1 | 22.9 | 8.5 | 17.1 | 18.3 |
| | VPT | 14.2 | 15.3 | 14.1 | 20.9 | 6.9 | 14.5 | 6.3 | 20.0 | 22.5 | 31.6 | 33.8 | 29.1 | 23.4 | 8.7 | 17.6 | 18.6 |
| | DPE | 14.1 | 15.4 | 13.9 | 22.5 | 6.6 | 14.8 | 6.4 | 20.3 | 23.2 | 33.1 | 34.6 | 29.9 | 23.8 | 8.5 | 17.4 | 19.0 |
| | Adapter | **14.3** | 15.7 | 15.0 | 22.4 | 9.2 | 15.6 | 6.7 | 20.9 | 24.0 | 33.0 | 34.3 | 30.7 | 25.3 | 10.2 | 20.2 | 19.8 |
| | **Our** | 14.1 | **16.1** | **15.4** | **23.1** | **9.7** | **15.8** | **7.7** | **21.6** | **24.7** | **33.1** | **36.4** | **30.9** | **25.6** | **11.8** | **20.9** | **20.5** |
| 4 | ZS | 5.1 | 4.9 | 4.4 | 14.6 | 4.0 | 8.1 | 4.4 | 15.7 | 22.1 | 32.7 | 33.6 | 19.0 | 19.7 | 4.3 | 9.1 | 13.4 |
| | TPT | 4.5 | 4.2 | 3.8 | 13.8 | 3.5 | 7.8 | 4.0 | 15.6 | 21.4 | 32.0 | 32.9 | 18.9 | 19.5 | 4.0 | 8.7 | 13.0 |
| | VPT | 5.7 | 5.6 | 5.0 | 13.9 | 4.6 | 8.0 | 4.5 | 16.0 | 21.8 | 31.8 | 32.6 | 19.0 | 19.5 | 4.8 | 9.8 | 13.5 |
| | DPE | 5.6 | 5.4 | 5.1 | 15.1 | 4.2 | 8.5 | 4.6 | 16.2 | 22.2 | 32.6 | 33.9 | 19.4 | 20.2 | 4.9 | 10.3 | 13.9 |
| | Adapter | 6.1 | 6.4 | 5.5 | 15.7 | 5.6 | 8.7 | 4.6 | 16.9 | 23.7 | 33.0 | 33.3 | 21.0 | **21.9** | 6.3 | 12.2 | 14.7 |
| | **Our** | **6.5** | **6.7** | **5.9** | **16.3** | **5.9** | **8.9** | **5.3** | **17.2** | **24.4** | **34.9** | **34.8** | **21.3** | 21.8 | **11.9** | **13.2** | **15.7** |
| 5 | ZS | 1.0 | 1.6 | 0.1 | 8.8 | 2.8 | 5.2 | 3.7 | 15.4 | 20.0 | 31.4 | 31.8 | 5.4 | 15.0 | 2.8 | 4.2 | 9.9 |
| | TPT | 0.8 | 1.4 | 0.7 | 8.2 | 2.3 | 4.7 | 3.3 | 14.9 | 19.2 | 30.8 | 31.1 | 5.9 | 15.2 | 2.6 | 4.0 | 9.7 |
| | VPT | 1.0 | 1.8 | 1.0 | 8.6 | 2.9 | 5.4 | 3.9 | 15.6 | 19.9 | 30.6 | 31.1 | 6.2 | 14.8 | 3.3 | 4.6 | 10.0 |
| | DPE | 1.1 | 1.9 | 0.8 | 9.1 | 2.9 | 5.6 | 4.2 | 15.7 | 20.3 | 31.1 | 32.3 | 5.9 | 15.8 | 3.2 | 5.2 | 10.3 |
| | Adapter | 1.1 | 2.1 | 1.0 | 9.8 | 3.2 | 6.0 | 4.0 | 16.8 | 21.4 | 32.1 | 31.7 | 7.6 | 16.7 | 4.1 | 6.2 | 10.9 |
| | **Our** | **1.9** | **2.3** | **1.7** | **10.4** | **3.4** | **6.9** | **5.1** | **16.9** | **21.7** | **33.2** | **32.1** | **7.9** | **17.1** | **4.7** | **6.7** | **11.5** |

Table 15: **Detection performance of our method compared against Zero-shot for all severity levels on COCO-C.** We report results for the **YOLO-World** detector on 15 different corruptions and five different severity levels. For each corruption, we present **mAP**. The best results are highlighted in bold.

with strong improvements on challenging digital transforms such as pixelate, JPEG, and contrast, and solid gains on classical noise corruptions. Overall, our method yields average improvements of 2.6 mAP and 2.6 $AP_{75}$ over ZS, indicating improved robustness across both metrics.

## A.16 QUALITATIVE ANALYSIS OF ENTROPY AND IoU-WEIGHTED ENTROPY

In Fig. 13, we compare standard entropy with IoU-weighted entropy. The heatmaps visualize clusters formed from the ZS proposals. The person cluster has a low maximum score (0.16) but is the largest (94 proposals), whereas the largest bird cluster is smaller (34 proposals). Standard entropy ignores this structure and uniformly sharpens proposals. Because there are more bird boxes overall, standard entropy tends to raise bird scores while the person class's score drops, leading to a false negative. In contrast, IoU-weighted entropy exploits the overlap structure, assigning greater weight to the largest coherent cluster (person) and less to smaller clusters (bird), thereby producing the correct detection.

## A.17 ADDITIONAL DETECTION VISUALIZATIONS

Fig. 14 provides additional qualitative examples from BDD, ExDark, Comic, and ClipArt. For each image, we show GT, ZS, Adapter, and VLOD-TTA. Compared with ZS and Adapter, our method typically (i) removes obvious false positives, (ii) recovers missed objects under low light and style shift, and (iii) yields tighter boxes with fewer duplicates. These trends match the quantitative gains reported in the main text.

| Sev. | Method | Noise | | | Blur | | | | Weather | | | Digital | | | | | Avg |
|---|---|---|---|---|---|---|---|---|---|---|---|---|---|---|---|---|---|
| | | Gaussi. | Shot | Impulse | Defocus | Glass | Motion | Zoom | Snow | Frost | Fog | Bright. | Contrast | Elastic | Pixel | JPEG | |
| 1 | ZS | 41.5 | 42.0 | 36.1 | 46.0 | 40.8 | 44.0 | 24.3 | 39.0 | 45.4 | 48.7 | 51.3 | 48.8 | 46.6 | 35.1 | 38.7 | 41.9 |
| | TPT | 38.9 | 39.5 | 33.5 | 43.5 | 41.8 | 43.3 | 22.9 | 28.5 | 44.4 | 48.0 | 50.7 | 48.1 | 45.7 | 34.5 | 38.0 | 40.1 |
| | VPT | 41.0 | 41.4 | 36.2 | 44.9 | 42.3 | 43.2 | 23.9 | 38.7 | 44.0 | 47.6 | 50.2 | 47.7 | 45.6 | 35.6 | 39.0 | 41.4 |
| | DPE | 41.5 | 41.7 | 36.6 | 46.2 | 41.3 | 44.5 | 24.9 | 39.6 | 45.8 | 48.8 | 51.1 | 48.9 | 46.9 | 35.4 | 39.3 | 42.2 |
| | Adapter | 41.4 | 42.1 | 37.0 | 45.9 | 42.3 | 44.7 | 25.2 | 40.1 | 45.9 | 48.8 | 50.7 | 48.8 | 47.2 | 40.2 | 41.7 | 42.8 |
| | **Our** | **42.1** | **43.1** | **37.1** | **47.6** | **42.5** | **44.9** | **25.5** | **40.7** | **46.6** | **49.7** | **52.4** | **49.5** | **47.5** | **41.9** | **42.5** | **43.6** |
| 2 | ZS | 33.4 | 33.2 | 27.6 | 41.9 | 31.3 | 35.1 | 16.7 | 28.7 | 38.1 | 47.6 | 50.2 | 46.6 | 42.2 | 24.1 | 30.3 | 35.1 |
| | TPT | 30.9 | 31.2 | 25.2 | 40.8 | 30.6 | 34.4 | 16.9 | 27.5 | 37.6 | 45.8 | 48.6 | 45.0 | 40.4 | 24.4 | 30.8 | 34.0 |
| | VPT | 33.6 | 33.6 | 28.1 | 40.7 | 32.0 | 34.6 | 16.5 | 28.7 | 37.7 | 46.4 | 49.1 | 45.5 | 41.4 | 25.5 | 30.8 | 34.9 |
| | DPE | 33.9 | 34.3 | 28.6 | 42.2 | 32.3 | 35.6 | 17.0 | 28.9 | 37.8 | 47.8 | 49.5 | 46.9 | 42.9 | 25.2 | 30.4 | 35.6 |
| | Adapter | 33.5 | 34.2 | 29.8 | 42.0 | **35.5** | 36.3 | 17.3 | 30.8 | **40.0** | 47.9 | 49.5 | 46.9 | **44.0** | 29.9 | 35.0 | 36.8 |
| | **Our** | **34.8** | **34.8** | **29.9** | **43.1** | **35.5** | **36.9** | **17.9** | **30.9** | **40.0** | **48.5** | **51.1** | **49.5** | 43.7 | **34.3** | **35.5** | **37.8** |
| 3 | ZS | 20.5 | 22.2 | 20.1 | 32.9 | 9.5 | 23.1 | 13.7 | 29.4 | 33.1 | 46.1 | 49.0 | 41.9 | 35.2 | 11.1 | 25.1 | 27.5 |
| | TPT | 19.7 | 21.3 | 19.1 | 32.0 | 8.3 | 22.5 | 13.9 | 28.2 | 32.6 | 45.6 | 48.4 | 41.7 | 34.8 | 11.7 | 25.6 | 27.0 |
| | VPT | 21.6 | 23.1 | 21.3 | 32.2 | 10.8 | 23.4 | 13.7 | 29.7 | 33.1 | 45.1 | 47.9 | 41.8 | 35.5 | 12.8 | 27.0 | 27.9 |
| | DPE | 20.8 | 22.9 | 21.1 | 33.4 | 10.3 | 23.8 | 14.0 | 29.9 | 33.3 | 46.3 | 48.6 | 42.3 | 35.9 | 12.5 | 26.4 | 28.1 |
| | Adapter | 22.2 | 23.8 | 22.6 | 33.6 | 14.5 | 25.0 | 14.5 | 30.9 | 34.8 | 46.4 | 50.5 | 43.4 | 38.4 | 14.7 | 30.6 | 29.6 |
| | **Our** | **22.8** | **24.3** | **22.9** | **34.1** | **14.9** | **25.1** | **14.8** | **31.2** | **35.6** | **47.4** | **50.5** | **43.5** | **38.5** | **17.0** | **31.3** | **30.3** |
| 4 | ZS | 7.8 | 7.4 | 6.7 | 22.6 | 6.1 | 13.4 | 10.1 | 23.5 | 32.0 | 45.9 | 47.3 | 27.5 | 30.2 | 6.2 | 14.0 | 20.0 |
| | TPT | 6.8 | 6.4 | 5.8 | 21.2 | 5.2 | 12.9 | 9.3 | 23.6 | 31.4 | 45.4 | 46.8 | 27.7 | 29.0 | 5.8 | 13.1 | 19.4 |
| | VPT | 8.7 | 8.6 | 7.7 | 22.2 | 7.1 | 13.5 | 10.4 | 24.1 | 31.9 | 45.2 | 46.4 | 28.0 | 30.1 | 7.1 | 15.1 | 20.4 |
| | DPE | 8.5 | 8.4 | 7.8 | 23.2 | 6.5 | 13.9 | 10.6 | 24.5 | 32.3 | 46.0 | 47.6 | 28.2 | 30.9 | 7.4 | 15.5 | 20.8 |
| | Adapter | **9.6** | 10.0 | 8.6 | 24.5 | 8.6 | 14.7 | 10.7 | **25.6** | 34.3 | 46.6 | 47.1 | **30.7** | 33.8 | 9.2 | 19.0 | 22.2 |
| | **Our** | 9.3 | **10.2** | **8.9** | **25.2** | **8.9** | 14.8 | **11.8** | **25.6** | **36.2** | **48.1** | **49.1** | **30.7** | **34.1** | **17.5** | **19.6** | **23.3** |
| 5 | ZS | 1.4 | 2.5 | 0.2 | 13.8 | 4.3 | 9.1 | 8.9 | 22.9 | 28.9 | 44.2 | 45.0 | 8.0 | 23.6 | 3.9 | 6.5 | 14.9 |
| | TPT | 1.2 | 2.1 | 1.1 | 12.8 | 3.5 | 8.1 | 8.1 | 21.6 | 37.2 | 43.8 | 44.4 | 8.2 | 24.0 | 3.6 | 6.6 | 15.1 |
| | VPT | 1.5 | 2.8 | 1.6 | 13.8 | 4.6 | 9.3 | 9.1 | 23.6 | 29.2 | 43.8 | 44.3 | 9.2 | 23.7 | 4.7 | 7.3 | 15.2 |
| | DPE | 1.6 | 2.9 | 1.4 | 14.2 | 4.5 | 9.5 | 9.3 | 23.4 | 29.6 | 44.1 | 45.6 | 8.9 | 24.8 | 4.4 | 7.9 | 15.5 |
| | Adapter | 1.6 | 3.3 | 1.6 | 15.9 | 5.1 | 10.5 | 9.4 | 25.2 | 31.3 | 45.2 | 45.0 | 11.4 | 26.5 | 5.9 | 9.8 | 16.5 |
| | **Our** | **1.9** | **3.4** | **1.9** | **16.4** | **5.5** | **10.9** | **9.8** | **25.2** | 31.4 | 45.9 | **46.1** | **11.7** | **26.7** | **6.3** | **9.9** | **16.9** |

Table 16: **Detection performance of our method compared against Zero-shot for all severity levels on COCO-C.** We report results for the **YOLO-World** detector on 15 different corruptions and five different severity levels. For each corruption, we present **mAP$_{50}$**. The best results are highlighted in bold.

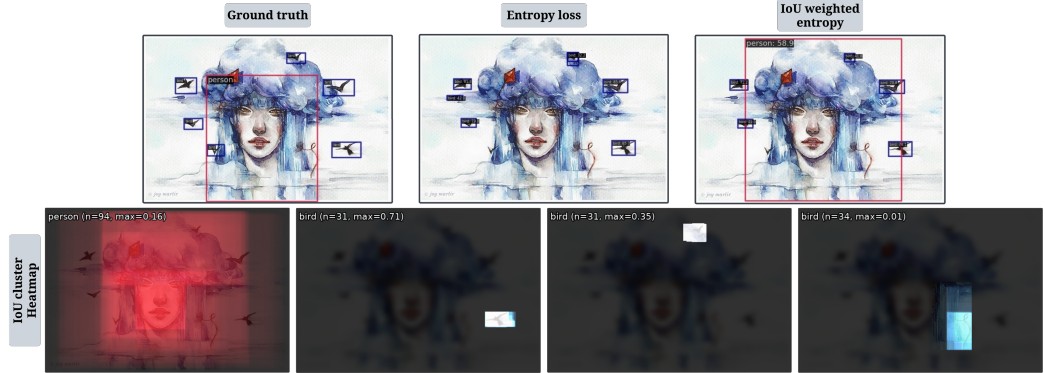

Figure 13: **Qualitative comparison of standard and IoU-weighted entropy.** Top row: ground truth and predictions using standard and IoU-weighted entropy. Bottom row: heatmaps of the IoU graph clusters from the ZS model. For each cluster, the predicted category, cluster size, and maximum score are displayed. Only the top four clusters are shown.

| Sev. | Method | Noise | | | Blur | | | | Weather | | | Digital | | | | | Avg |
|---|---|---|---|---|---|---|---|---|---|---|---|---|---|---|---|---|---|
| | | Gaussi. | Shot | Impulse | Defocus | Glass | Motion | Zoom | Snow | Frost | Fog | Bright. | Contrast | Elastic | Pixel | JPEG | |
| 1 | ZS | 31.3 | 31.5 | 27.1 | 35.3 | 30.5 | 32.5 | 13.3 | 29.4 | 34.9 | 37.8 | 39.9 | 37.9 | 34.8 | 26.8 | 28.6 | 31.4 |
| | TPT | 29.1 | 29.6 | 25.0 | 33.5 | 31.1 | 31.9 | 12.4 | 28.9 | 33.8 | 36.5 | 38.9 | 37.1 | 34.0 | 26.5 | 28.3 | 30.4 |
| | VPT | 30.2 | 30.5 | 26.0 | 33.2 | 30.2 | 31.1 | 12.7 | 28.7 | 33.2 | 35.7 | 38.1 | 35.7 | 33.9 | 26.6 | 28.5 | 30.3 |
| | DPE | 30.9 | 31.3 | 27.2 | 35.4 | 30.9 | 32.7 | 13.6 | 29.7 | 35.1 | 37.7 | 39.6 | 38.0 | 34.9 | 27.1 | 29.1 | 31.5 |
| | Adapter | 30.6 | 31.1 | 27.1 | 34.9 | 31.4 | 32.7 | 13.7 | 29.9 | 34.8 | 37.4 | 38.9 | 37.5 | 35.2 | 30.6 | 30.5 | 31.8 |
| | **Our** | **32.0** | **31.9** | **27.3** | **36.9** | **31.5** | **32.9** | **14.8** | **30.6** | **35.6** | **38.5** | **41.1** | **39.1** | **35.9** | **31.8** | **30.8** | **32.7** |
| 2 | ZS | 24.6 | 24.5 | 20.2 | 31.6 | 23.3 | 24.5 | 7.5 | 21.0 | 28.5 | 36.7 | 39.0 | 35.8 | 30.8 | 18.3 | 21.9 | 25.9 |
| | TPT | 22.9 | 23.1 | 18.4 | 31.1 | 23.1 | 24.3 | 7.9 | 20.7 | 28.0 | 35.6 | 38.1 | 35.1 | 29.9 | 18.6 | 21.9 | 25.2 |
| | VPT | 24.1 | 24.1 | 20.0 | 29.6 | 23.3 | 23.5 | 7.0 | 20.6 | 27.2 | 34.6 | 37.1 | 34.3 | 29.9 | 19.0 | 21.8 | 25.1 |
| | DPE | 24.5 | 24.6 | 20.7 | 31.7 | 23.8 | 24.8 | 7.9 | 21.1 | 28.3 | 36.8 | 38.6 | 35.6 | 31.4 | 18.8 | 21.9 | 26.0 |
| | Adapter | 24.0 | 24.7 | 21.2 | 31.2 | 26.2 | 24.7 | 7.4 | 22.2 | **29.7** | 36.5 | 37.9 | 35.7 | **32.1** | 22.5 | 25.0 | 26.7 |
| | **Our** | **25.1** | **25.1** | **23.4** | **33.1** | **26.4** | **25.9** | **8.8** | **22.4** | **29.7** | **37.5** | **40.3** | **37.7** | 31.9 | **25.6** | **25.3** | **27.9** |
| 3 | ZS | 14.6 | 15.9 | 14.3 | 23.5 | 6.5 | 15.1 | 4.8 | 21.4 | 24.6 | 35.6 | 37.6 | 31.9 | 24.9 | 8.1 | 17.7 | 19.8 |
| | TPT | 14.1 | 14.5 | 14.0 | 23.1 | 5.6 | 15.1 | 5.1 | 20.9 | 23.8 | 34.4 | 36.8 | 31.2 | 24.1 | 8.6 | 18.0 | 19.3 |
| | VPT | 15.0 | 16.2 | 14.8 | 22.0 | 7.1 | 14.9 | 4.8 | 21.1 | 23.8 | 33.7 | 36.1 | 31.2 | 24.8 | 9.3 | 18.2 | 19.5 |
| | DPE | 14.9 | 16.2 | 14.7 | 24.1 | 7.4 | 15.4 | 5.0 | 21.9 | 24.7 | 35.9 | 37.5 | 31.9 | 25.3 | 9.2 | 18.3 | 20.2 |
| | Adapter | 15.5 | 16.4 | 15.7 | 23.8 | 10.0 | 16.1 | 5.2 | 22.2 | 25.7 | 35.4 | 36.7 | 32.8 | 26.3 | 10.8 | 21.0 | 20.9 |
| | **Our** | **15.8** | **16.9** | **16.3** | **24.4** | **10.2** | **16.5** | **6.0** | **22.6** | **26.1** | **37.4** | **38.9** | **32.9** | **26.4** | **12.6** | **21.8** | **21.7** |
| 4 | ZS | 5.4 | 5.2 | 4.4 | 15.5 | 4.2 | 8.4 | 3.1 | 16.6 | 23.7 | 35.2 | 36.1 | 20.5 | 20.6 | 4.6 | 9.6 | 14.2 |
| | TPT | 4.7 | 4.4 | 4.0 | 14.7 | 3.7 | 8.1 | 2.8 | 16.6 | 22.9 | 34.6 | 35.3 | 20.2 | 20.5 | 4.3 | 9.2 | 13.7 |
| | VPT | 5.9 | 5.8 | 5.1 | 14.5 | 4.7 | 8.1 | 3.1 | 16.8 | 23.3 | 34.2 | 34.7 | 20.3 | 20.5 | 5.1 | 10.3 | 14.2 |
| | DPE | 5.7 | 5.7 | 5.1 | 15.2 | 4.3 | 8.9 | 3.3 | 16.9 | 23.7 | 35.4 | 36.4 | 20.8 | 21.1 | 5.3 | 10.5 | 14.6 |
| | Adapter | 6.3 | 6.7 | 5.8 | 16.5 | 5.8 | 8.8 | 3.1 | 17.9 | 25.2 | 35.2 | 35.6 | 22.5 | **22.9** | 6.7 | 12.8 | 15.5 |
| | **Our** | **6.9** | **6.8** | **6.2** | **16.8** | **6.2** | **9.8** | **5.8** | **18.7** | **25.6** | **37.1** | **37.3** | 22.2 | 22.7 | **12.7** | **13.1** | **16.5** |
| 5 | ZS | 0.9 | 1.6 | 0.1 | 9.3 | 2.9 | 5.3 | 2.6 | 16.4 | 21.3 | 33.8 | 34.1 | 5.8 | 15.4 | 3.0 | 4.4 | 10.5 |
| | TPT | 0.8 | 1.4 | 0.7 | 8.7 | 2.5 | 4.8 | 2.3 | 15.8 | 20.5 | 33.2 | 33.4 | 6.2 | 15.8 | 2.8 | 4.5 | 10.2 |
| | VPT | 1.0 | 1.8 | 1.1 | 9.0 | 3.0 | 5.3 | 2.7 | 16.4 | 21.0 | 32.6 | 32.9 | 6.5 | 15.2 | 3.5 | 4.8 | 10.5 |
| | DPE | 1.0 | 1.8 | 0.8 | 9.4 | 3.2 | 5.5 | 3.1 | 16.8 | 21.8 | 33.6 | 34.3 | 6.4 | 16.0 | 3.3 | 5.1 | 10.8 |
| | Adapter | 1.1 | **2.2** | 1.0 | 10.1 | 3.3 | 5.8 | 2.8 | 17.6 | 22.6 | 34.5 | 33.8 | **7.9** | 17.1 | 4.3 | 6.5 | 11.4 |
| | **Our** | **1.3** | **2.2** | **1.6** | **10.6** | **3.5** | **6.4** | **4.2** | **17.8** | **23.0** | **35.1** | **35.4** | 7.4 | **17.3** | **4.9** | **7.3** | **11.9** |

Table 17: **Detection performance of our method compared against Zero-shot for all severity levels on COCO-C.** We report results for the **YOLO-World** detector on 15 different corruptions and five different severity levels. For each corruption, we present $\text{mAP}_{75}$. The best results are highlighted in bold.

| Method | Noise | | | Blur | | | | Weather | | | Digital | | | | | Avg |
|---|---|---|---|---|---|---|---|---|---|---|---|---|---|---|---|---|
| | Gaussi. | Shot | Impulse | Defocus | Glass | Motion | Zoom | Snow | Frost | Fog | Bright. | Contrast | Elastic | Pixel | JPEG | |
| ZS | 7.2 | 7.1 | 6.8 | 35.8 | 10.3 | 22.4 | 12.6 | 24.3 | 36.1 | 54.9 | 57.4 | 31.5 | 36.9 | 6.8 | 14.0 | 24.3 |
| TPT | 7.5 | 7.9 | 7.2 | 36.1 | 10.4 | 22.1 | 12.8 | 25.1 | 36.8 | 54.7 | 57.5 | 32.2 | 37.2 | 7.2 | 15.1 | 24.7 |
| VPT | 7.9 | 7.6 | 7.1 | 35.6 | 10.8 | 22.1 | 12.6 | 24.8 | 36.4 | 54.9 | 56.8 | 32.7 | 37.3 | 7.4 | 15.3 | 24.6 |
| DPE | 7.9 | 8.2 | 7.6 | 36.3 | 10.9 | 22.6 | 12.8 | 25.3 | 36.9 | 55.1 | 57.9 | 33.6 | 37.4 | 8.2 | 16.4 | 25.1 |
| Adapter | 9.0 | 9.5 | 8.1 | 36.4 | 12.9 | 21.5 | 11.9 | 26.7 | 37.5 | 54.3 | 55.5 | 33.4 | 40.1 | 10.1 | 19.2 | 25.7 |
| **Our** | **10.3** | **9.9** | **9.2** | **38.1** | **14.9** | **22.7** | **12.9** | **27.2** | **38.4** | **55.6** | **58.1** | **35.4** | **40.7** | **10.6** | **19.5** | **26.9** |

Table 18: **Detection performance of different test-time adaptation strategies on PASCAL-C.** We report results for the **YOLO-World** detector on 15 different corruptions. For each corruption, we present **mAP**. The best results are highlighted in bold.

| Method | Noise | | | Blur | | | | Weather | | | Digital | | | | | Avg |
|---|---|---|---|---|---|---|---|---|---|---|---|---|---|---|---|---|
| | Gaussi. | Shot | Impulse | Defocus | Glass | Motion | Zoom | Snow | Frost | Fog | Bright. | Contrast | Elastic | Pixel | JPEG | |
| ZS | 7.4 | 7.1 | 6.9 | 39.3 | 10.8 | 24.1 | 8.7 | 26.2 | 39.1 | 59.8 | 62.8 | 34.1 | 40.2 | 7.1 | 15.0 | 25.9 |
| TPT | 7.9 | 7.8 | 7.2 | 39.4 | 11.2 | 24.0 | 9.0 | 26.9 | 39.3 | 59.7 | 62.9 | 35.2 | 41.1 | 7.9 | 16.2 | 26.4 |
| VPT | 7.9 | 7.7 | 7.2 | 38.5 | 11.4 | 23.5 | 8.7 | 26.2 | 39.3 | 59.8 | 61.8 | 35.3 | 41.2 | 7.8 | 16.4 | 26.2 |
| DPE | 8.3 | 8.3 | 7.9 | 39.5 | 11.3 | 24.3 | 9.1 | 27.2 | 39.8 | 59.9 | 63.1 | 35.9 | 41.5 | 8.3 | 17.6 | 26.8 |
| Adapter | 9.1 | 9.7 | 8.3 | 39.2 | 13.6 | 22.6 | 8.3 | 28.0 | 40.4 | 58.9 | 60.2 | 35.7 | 43.3 | 10.7 | 20.5 | 27.2 |
| **Our** | **10.3** | **10.2** | **9.1** | **40.9** | **15.4** | **24.4** | **8.9** | **28.8** | **41.5** | **60.1** | **63.3** | **38.1** | **44.3** | **11.1** | **20.6** | **28.5** |

Table 19: **Detection performance of different test-time adaptation strategies on PASCAL-C.** We report results for the **YOLO-World** detector on 15 different corruptions. For each corruption, we present $\text{AP}_{75}$. The best results are highlighted in bold.

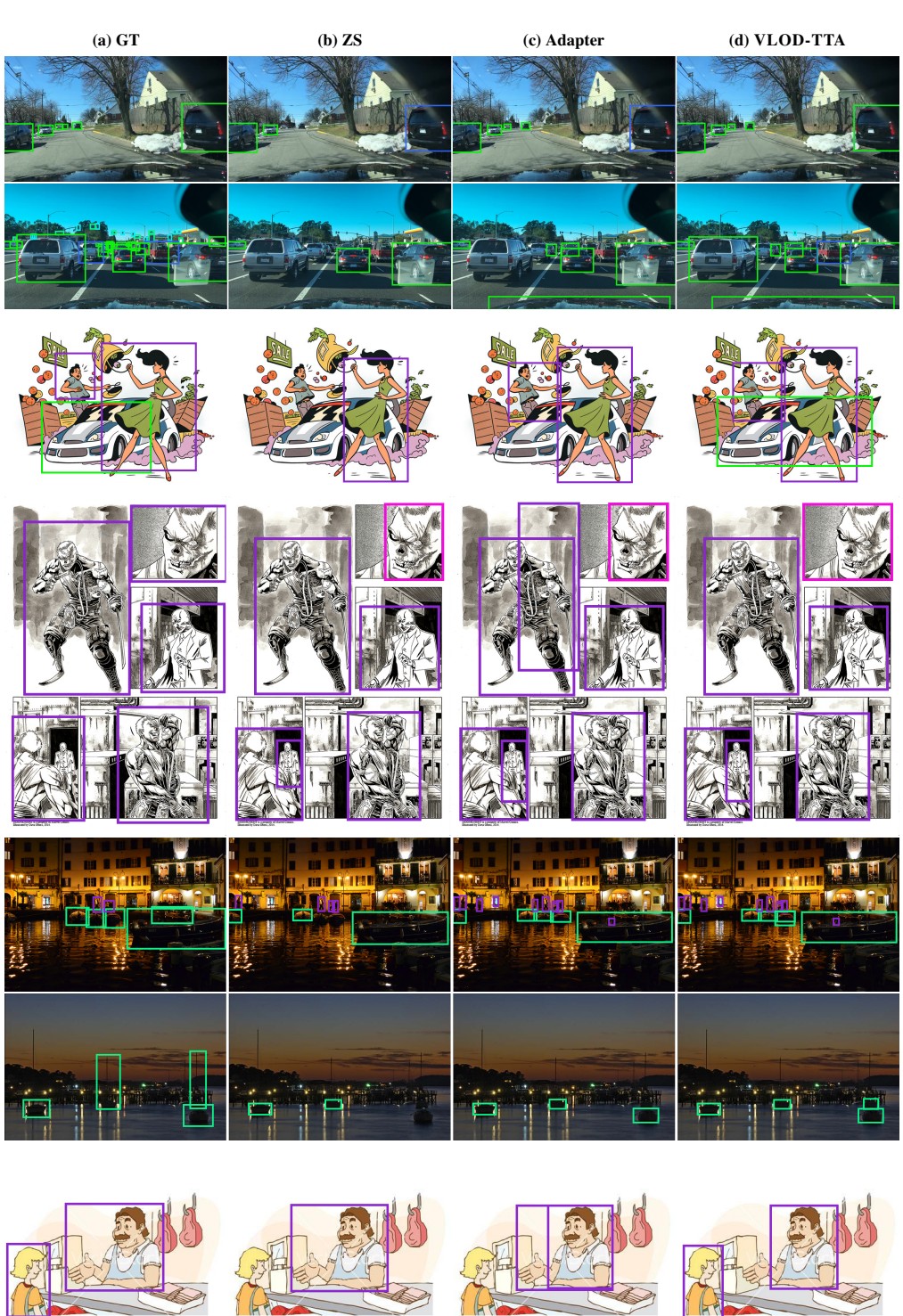

Figure 14: **YOLO-World detections across different approaches**: Each column corresponds to a different approach: (a) GT (Ground Truth), (b) ZS (Zero-Shot), (c) Adapter, and (d) VLOD-TTA.

