# OpenReview forum: "VLOD-TTA: Test-Time Adaptation of Vision-Language Object Detectors"
_ICLR.cc/2026/Conference — Submitted to ICLR 2026_

### Official Review · Reviewer_2uSN · 2025-10-28

**Soundness:** 3
**Presentation:** 3
**Contribution:** 3
**Rating:** 4
**Confidence:** 3

**Summary:**

This paper proposes VLOD-TTA, the first test-time adaptation (TTA) framework specifically designed for vision-language object detectors (VLODs). Its core goal is to address the significant performance degradation of VLODs when faced with new environments that differ from the training data distribution (e.g., "domain drift" due to changes in style, lighting, and weather).

**Strengths:**

1.The paper establishes a comprehensive benchmark, validating the method’s effectiveness across up to 96 different test scenarios. The results show that VLOD-TTA consistently improves model performance under various domain shifts, including artistic styles, real-world driving scenes, and image corruptions.
2.The paper concerns DA problem by tta, which is easy to follow

**Weaknesses:**

[1] The IWE mechanism relies on clusters of candidate boxes with dense overlaps. Therefore, when dealing with a large number of tiny, sparse, and low-overlap objects—such as in scenes from the Cityscapes dataset—its effectiveness may be reduced, as forming sufficiently “high-density” regions to guide optimization becomes challenging.
[2] The construction of this prompt pool (generated using GPT in the paper) is itself a labor-intensive step. Moreover, for highly specialized domains, the existing prompt pool may be insufficiently comprehensive, which could limit the effectiveness of IPS.

**Questions:**

[1] TTA introduces significant latency. How can we balance performance gains with computational cost? In the future, it may be worth exploring a **“selective adaptation”** strategy, where the model first evaluates the degree of domain shift in the current input and only triggers the TTA process when the shift exceeds a certain threshold, thereby maintaining fast inference in most cases.
[2] VLOD-TTA demonstrates effectiveness under various domain shifts. However, is there a “limit” to its adaptation capability?

---

> ### Author Response · Authors · 2025-11-21
> **Response to Reviewer 2uSN [1/2]**
>
> Thank you for the insightful comments. Next, we address your concerns and highlight the corresponding changes in the updated manuscript.
>
> > ### **W1. Overcoming IWE underperformance on the Cityscapes dataset**
>
> We agree that IoU-weighted entropy (IWE) is most effective when objects induce clusters of overlapping proposals, which is harder for tiny, sparse objects. In Cityscapes, this issue is amplified by resolution. The dataset contains many small objects at the original 2048×1024 resolution. YOLO-World resizes inputs to 640×640, which further shrinks distant cars and pedestrians and often causes missed detections. At this resolution, many small objects do not generate stable overlapping proposals, so IWE underperforms.
> To analyze this, we re-evaluate with a larger input size of 1024×1024. As shown in the table below, the zero-shot baseline increases substantially with the increase in image resolution. VLOD-TTA then improves further, with a modest gain at 640 (+0.6 mAP) but a much larger gain at 1024 (+2.7 mAP). At higher resolution, more small objects produce stable overlapping proposals that IWE can weight and sharpen during adaptation. This supports the reviewer’s concern for tiny, sparse objects. It also indicates that the underperformance on Cityscapes is mainly due to small objects and aggressive downsampling rather than an inherent limitation of the IWE mechanism itself. We have added the visualizations and ablations for an in-depth analysis of the Cityscapes underperformance in Appendix A2 of the revised manuscript.
>
> | Cityscapes     | mAP  | mAP50 | mAP75 |
> | -------------- | ---- | ----- | ----- |
> | Zero-shot (640)            | 18.8 | 31.0  | 17.9  |
> | **VLOD-TTA (640)**          | 19.4 | 31.8  | 18.6  |
> | Zero-shot (1024)  | 28.5 | 43.9  | 28.3  |
> | **VLOD-TTA (1024)** | **31.2** | **46.6**  | **30.5**  |
>
> > ### **W2.1 Generation of prompt pool being labour-intensive**
>
> In our setup, prompt generation is simple and straightforward. For any custom dataset, we query a large language model (ChatGPT-5) with “Generate 16 prompts for each object category: {*category list*}”. This is a one-shot step with only a brief sanity check, so the required human effort is negligible compared with model training and adaptation.
>
> > ### **W2.2 IPS effectiveness for specialized domains**
>
> This is a very relevant concern. To examine it, we ran an experiment on the Aquarium Object Detection dataset [1], which contains underwater animals with categories such as fish, jellyfish, penguin, puffin, shark, starfish, and stingray. These fine-grained aquatic categories are rare in generic benchmarks and therefore constitute a specialized and challenging domain for IPS.
> The table below shows that CLIP-Style Prompts degrade the zero-shot (ZS) performance. In contrast, IPS improves performance over the ZS baseline, and combining IPS with IWE (VLOD-TTA) yields further gain. This indicates that both IPS and IWE remain effective even in specialized domains. We have added the experiment on the Aquarium Object Detection dataset in Appendix A13 of the updated manuscript.
>
> | Aquarium | mAP  | mAP50 | mAP75 |
> | -------- | ---- | ----- | ----- |
> | Zero-shot       | 11.9 | 20.3  | 11.6  |
> | CLIP-Style Prompts       | 11.6 | 19.7  | 11.2  |
> | IPS      | 12.4 | 21.5  | 12.1  |
> | **VLOD-TTA** | **14.5** | **25.1**  | **14.8**  |
>
>
> > ### **Q1. Balancing computation cost with selective adaptation strategy**
>
> Inference-time complexity is a key challenge for TTA methods. A selective adaptation strategy is indeed an interesting direction and has already been explored for conventional object detectors by triggering adaptation only when the distance to source statistics becomes large [2]. Extending this idea to VLODs is more challenging. They are trained on massive and heterogeneous source corpora, which makes computing reliable source statistics nontrivial. We therefore view selective adaptation for VLODs as an important avenue for future work. Additionally, backpropagation-free TTA methods inspired by recent cache-based approaches for VLMs such as TDA [3] would further reduce latency.

---

> ### Author Response · Authors · 2025-11-21
> **Response to Reviewer 2uSN [2/2]**
>
> > ### **Q2. Is there a limit to VLOD-TTA adaptation capability?**
>
> Yes, there is a limit. Similar to other TTA methods, VLOD-TTA can only refine what is already present in the pretrained OD and its zero-shot predictions. If the domain shift is extreme, categories are essentially unseen, or the initial proposals are very poor, the gains will be small. For the COCO-C dataset (see Appendix A14) across severity levels, gains are most pronounced at severities 2–4. At severity 1, the improvements are smaller, as the test distribution remains close to the training distribution. At severity 5, performance is severely degraded for all methods, making most predictions unreliable, so it is challenging to improve. In practice, VLOD-TTA works best when the zero-shot detector still produces partially meaningful boxes and scores, and its benefits diminish once that assumption breaks.
>
> ### **References**
>
> *[1] Roboflow. Aquarium Object Detection Dataset. Roboflow Universe, 2020. Available at: https://public.roboflow.com/object-detection/aquarium.*
>
> *[2] Yoo, J., Lee, D., Chung, I., Kim, D., & Kwak, N. (2024). What how and when should object detectors update in continually changing test domains?. In Proceedings of the IEEE/CVF Conference on Computer Vision and Pattern Recognition (pp. 23354-23363).*
>
> *[3] Karmanov, A., Guan, D., Lu, S., El Saddik, A., and Xing, E. “Efficient Test-Time Adaptation of Vision-Language Models.” In Proceedings of the IEEE/CVF Conference on Computer Vision and Pattern Recognition (CVPR), 2024.*

---

### Official Review · Reviewer_YXiS · 2025-10-29

**Soundness:** 3
**Presentation:** 3
**Contribution:** 2
**Rating:** 4
**Confidence:** 3

**Summary:**

This paper proposes VLOD-TTA, a test-time adaptation framework for vision-language object detectors such as YOLO-World and Grounding DINO. The approach combines IoU-weighted entropy minimization to focus adaptation on spatially coherent proposal clusters and image-conditioned prompt selection to fuse only the most relevant prompts with detector outputs. Only lightweight adapters are optimized during test time. Experiments across several domain shifts indicate consistent improvements over zero-shot and baseline TTA strategies.

**Strengths:**

- **Relevant problem**.

    Test-time adaptation for open-vocabulary object detection remains underexplored and has strong relevance for real-world robustness.


- **Solid empirical results**.

    The study covers multiple datasets and two state-of-the-art VLODs, showing consistent gains.

**Weaknesses:**

- **Architecture-dependent adaptation**.

    Different parameters are adapted for YOLO-World and Grounding DINO, which reduces generality and complicates baseline comparisons.

- **Unclear adaptation protocol**.

    It is not specified whether the model is reset after each image or adapts continuously, which can lead to very different behavior and raises reproducibility concerns.

- **Unusual adaptation target**.

    The method updates lightweight adapters rather than normalization parameters, which differs from standard TTA practice. The rationale for this choice should be clarified.

- **Significance of the contributions**.

    The proposed image-conditioned prompt selection and IoU-weighted entropy minimization modules seem reasonable but somewhat limited in scope. While they contribute to performance it is not clear that these represent sufficiently substantial innovations to justify a top-tier venue.

**Questions:**

1.	Are the model parameters reset after every image, or does adaptation accumulate across the evaluation set? Please clarify the protocol and its practical justification.
2.	Could the IoU-weighted entropy component be used alone without depending on adapters, or could adapters be placed uniformly at shallow layers to avoid architecture-specific tuning?
3.	Did you evaluate or consider updating BN or LayerNorm statistics? How would a normalization-only adaptation compare in terms of accuracy and compute?

---

> ### Author Response · Authors · 2025-11-21
> **Response to Reviewer YXiS [1/3]**
>
> Thank you for the thorough review. Next, we address your concerns and highlight the changes in the updated manuscript.
>
>
> > ### **W1. Architecture-dependent adaptation**
>
> We agree that the adapted parameters differ across YOLO-World and Grounding DINO, but this is a consequence of their very different architectures and pre-training rather than a lack of generality. In YOLO-World, the detector and text encoder are treated as decoupled blocks and interact only through cosine similarity at the final scoring stage. Updating the text encoder therefore mainly changes classification scores and yields limited gains. This is consistent with the YOLO-World original paper's pre-training result that fine-tuning the text encoder can hurt performance [1]. So we follow that design and adapt only vision-side adapters. Grounding DINO instead uses early cross-modal fusion, where the text encoder is tightly integrated into detection, so adapting the text encoder can directly influence both localization and classification. During pretraining, the original paper reports that optimizing the text encoder jointly with the vision backbone results in better performance [2], so adapting text-side adapters is the natural choice. Grounding DINO tends to overfit when optimizing the vision encoder on a single image, due to its much heavier architecture (172M parameters for Grounding DINO vs 13M for YOLO-World).
>
> Our contribution lies in the IoU-weighted entropy loss and the prompt selection strategy, which are shared across both architectures. For each architecture, we include an adapter baseline that adapts the same parameters using standard entropy, which assures fair comparison. This shows that our gains come from the proposed ideas rather than from a more favorable choice of adapted layers. We have added a detailed theoretical justification of why we optimize different parameters for each architecture in Section 3.4 (Model Update) of the updated manuscript.
>
> > ### **W3, Q3. Can batch normalization parameters be trained instead of adapters? If yes, why are they not trained, and what is the complexity of this alternative?**
>
> Yes, we did consider batch norm statistics for adaptation and have added an ablation comparing batch norm and adapters in Section 4.3 (Ablation Studies) of the revised manuscript. The table below reports four settings. “Adapters” and “B. Norm” correspond to adapting adapter and batch norm parameters, respectively, with the standard entropy loss. “VLOD-TTA” and “B. Norm VLOD-TTA” use our proposed loss while adapting adapter and batch norm parameters, respectively. Batch norm with our loss reaches almost the same performance as the adapter baseline, which shows that batch norm parameters can be used instead of adapter parameters.
>
> **Why optimize adapters instead of batch norm?** With adapters, the best performance is obtained across datasets with a single learning rate of 5e-3. For batch norm, different learning rates are required to reach the best performance for each dataset, for example, 1e-2 for Watercolor and 3e-2 for ClipArt. In the TTA setting, the target domain is unknown at test-time, so tuning the learning rate per-dataset is impractical. For this reason, we choose to update adapter parameters in our main experiments.
>
> **Complexity Comparison with Batch Norm -** Batch norm has 0.021M trainable parameters and an FPS of 21, while adapters have 1.61M trainable parameters and an FPS of 20. In terms of memory, batch norm is lighter and the FPS is almost identical for both. However, batch norm requires fine-tuning the learning rate for each dataset to achieve a high level of performance, which reduces its practical applicability.
>
> | Method          |      | Water |       |      | ClipArt |       |      | Comic |       |
> | :-------------- | :--: | :--------: | :---: | :--: | :-----: | :---: | :--: | :---: | :---: |
> |                 |  mAP |    mAP50   | mAP75 |  mAP |  mAP50  | mAP75 |  mAP | mAP50 | mAP75 |
> | Zero-shot       | 26.9 |    47.9    |  25.9 | 24.4 |   40.1  |  26.3 | 17.8 |  29.4 |  18.8 |
> | B.Norm          | 28.4 |    51.3    |  26.5 | 26.7 |   44.1  |  28.1 | 20.6 |  34.5 |  21.7 |
> | Adapters        | 28.3 |    51.5    |  26.7 | 26.9 |   44.1  |  27.8 | 20.8 |  34.7 |  21.7 |
> | **B.Norm VLOD-TTA** | 29.4 |    52.9    |  **28.8** | **28.3** |   45.3  |  **30.0** | 21.3 |  **36.1** |  22.0 |
> | **VLOD-TTA**        | **29.6** |    **53.1**    |  28.7 | 28.1 |   **45.4**  |  29.9 | **21.4** |  **36.1** |  **22.1** |

---

> ### Author Response · Authors · 2025-11-21
> **Response to Reviewer YXiS [2/3]**
>
> > ### **W2, Q1. Do the parameters reset after each update?**
>
> Our work follows an episodic TTA protocol [4] rather than a continual TTA protocol [5]. As described in Section 3.4 (Model Update), the adapted parameters are reset to their zero-initialized values after each image. Concretely, for each test image, we optimize the adapters, generate the prediction, and then reset the adapters before moving to the next image. Because the model is optimized on a single image, the adapted parameters are not reliable for other images, and accumulating them across the evaluation set can cause drift and hurt performance. Episodic TTA is also better suited to changing environmental conditions, since adaptation does not depend on the order of images or past states. Exploring continual TTA is interesting future work. We have added more details on the protocol in Section 3.4 (Model Update) of the revised manuscript.
>
>
> > ### **W4. Significance of the contributions**
>
> Our contribution lies in the IoU-weighted entropy loss and the image-conditioned prompt selection strategy, which together define a TTA framework tailored to VLODs.
>
> **IoU-weighted entropy -** Entropy minimization [3] is one of the most effective TTA losses, but it ignores the localization component in object detection. Our IoU-weighted entropy addresses this by focusing adaptation on spatially coherent regions. To further demonstrate the applicability of our IoU-weighting beyond entropy minimization, we integrate it into a standard pseudo-label-based TTA scheme for ODs. Pseudo-labeling is a widely used TTA strategy with a mean-teacher setup, where the teacher generates pseudo labels that supervise student updates. To obtain IoU-weighted pseudo labels, we cluster overlapping teacher boxes by IoU and weight each pseudo label by its normalized cluster size. The table below compares standard pseudo-labeling with its IoU-weighted variant on the Watercolor, ClipArt, and Comic datasets. IoU-weighted pseudo-labeling consistently improves mAP, mAP50, and mAP75 across all three datasets. This shows that IoU-based weighting can be plugged into existing pseudo-label losses rather than being restricted to our entropy-minimization objective. We have added this ablation study with the table below in Appendix A3 of the revised manuscript to document the IoU-weighted pseudo-labeling results in detail.
>
> | Method          |      | Water |       |      | ClipArt |       |      | Comic |       |
> | :-------------- | :--: | :--------: | :---: | :--: | :-----: | :---: | :--: | :---: | :---: |
> |                 |  mAP |    mAP50   | mAP75 |  mAP |  mAP50  | mAP75 |  mAP | mAP50 | mAP75 |
> | Zero-shot                                 | 26.9           | 47.9             | 25.9             | 24.4        | 40.1          | 26.3          | 17.8      | 29.4        | 18.8        |
> | Pseudo-label                       | 28.3           | 50.1             | 27.0             | 25.9        | 42.3          | 28.1          | 19.2      | 31.9        | 20.1        |
> | IoU-weighted Pseudo-label  | 29.1           | 51.6             | 27.9             | 27.3        | 43.7          | 29.0          | 20.5      | 33.2        | 21.2        |
>
> **Image-conditioned prompt selection -** Prompt averaging is an efficient strategy to improve the performance of VLMs, yet it underperforms for detection (see Appendix A7). To overcome this, we propose a prompt selection strategy that selects the prompt that works best for the current image. Our prompt-selection strategy is not limited to TTA and, similar to CLIP’s prompt averaging, can be used with any VLOD setup without additional computational complexity.

---

> ### Author Response · Authors · 2025-11-21
> **Response to Reviewer YXiS [3/3]**
>
> > ### **Q2. Could IoU-weighted entropy be used without adapters, or could adapters be placed uniformly?**
>
> Yes. IoU-weighted entropy is independent of adapters and can be applied to any trainable parameters, including batch norm alone. In our experiments, we place adapters only in a subset of layers to reduce computational cost. For YOLO-World, adapters can be placed uniformly in the vision module, while for Grounding DINO they can be placed in the text module. This variation is solely due to the architectural and pre-training differences between YOLO-World and Grounding DINO, and is not related to our proposed method.
>
> ### **References**
>
> *[1] Cheng, T., Song, L., Ge, Y., Liu, W., Wang, X., & Shan, Y. (2024). Yolo-world: Real-time open-vocabulary object detection. In Proceedings of the IEEE/CVF conference on computer vision and pattern recognition (pp. 16901-16911).*
>
> *[2] Liu, S., Zeng, Z., Ren, T., Li, F., Zhang, H., Yang, J., ... & Zhang, L. (2024, September). Grounding dino: Marrying dino with grounded pre-training for open-set object detection. In European conference on computer vision (pp. 38-55). Cham: Springer Nature Switzerland.*
>
> *[3] Wang, D., Shelhamer, E., Liu, S., Olshausen, B., & Darrell, T. Tent: Fully Test-Time Adaptation by Entropy Minimization. In International Conference on Learning Representations. 2021.*
>
> *[4] Shuai Zhao, Xiaohan Wang, Linchao Zhu, and Yi Yang. “Test-Time Adaptation with CLIP Reward for Zero-Shot Generalization in Vision-Language Models.” International Conference on Learning Representations (ICLR), 2024.*
>
> *[5] Cohen, N., Kahana, A., Adi, Y., and Michaeli, T. “COTTA: Continual Test-Time Adaptation.” In Proceedings of the IEEE/CVF Conference on Computer Vision and Pattern Recognition (CVPR), 2022.*

---

### Official Review · Reviewer_vTge · 2025-11-02

**Soundness:** 2
**Presentation:** 3
**Contribution:** 3
**Rating:** 6
**Confidence:** 4

**Summary:**

This paper introduces an approach to perform Test-Time Adaptation (TTA) for the task of Object Detection with Vision Language Models (VLM). The method is based on two main contributions: a weighting of the visual proposals contribution based on their Intersection over Union (IoU) weighting, and a prompt selection conditioned over the image. Extensive evaluations were performed on multiple datasets and corruptions.

**Strengths:**

The paper tackles TTA for object detection using VLM, which is an original and interesting approach.  The mIoU weighting of the entropy is simple yet original and well in phase is already existing TTA approaches.

The results show consistent gains in mAP on all datasets, which evaluate different kinds of simulated and natural corruption/domain shift.

**Weaknesses:**

TTA is generally conducted by optimizing batch norm parameters. Here, the method relies on adapters and learnable residual prompts. This parameter overhead might be enough to explain the gains of the proposed approach. Furthermore, if adaptation requires back-prop or large memory, it may not be viable for streaming/inference environments. This point should be discussed, and the number of learnable parameters and complexity should be shown for the method and the baselines.

 The paper lacks an ablation to disentangle the contribution of the mIoU weighting scheme and the prompt selection. Furthermore, more description would be required to understand how the prompts were generated using a GPT model, as this step appears to have a great impact.

**Questions:**

Which GPT model was used to generate the prompts?

Which prompts were given to the GPT model to construct the textual prompts?


It is unclear how the method deals with the different samples in the batch and how this parameter should have an impact on the TTA task.

---

> ### Author Response · Authors · 2025-11-21
> **Response to Reviewer vTge [1/2]**
>
> The authors are grateful for your positive feedback. Below, we address your concerns and highlight the changes we have made in the OpenReview submission to incorporate your suggestions.
>
>
> > ### **W1.1 Do gains originate only from additional adapter and residual parameters?**
>
>
> The gains are not solely due to adding extra parameters. To verify this, we removed the adapter parameters from the network and adapted only the batch norm layer parameters. The table below reports four settings. “Adapters” and “B. Norm” correspond to adapting adapter and batch norm parameters, respectively, with the standard entropy loss. “VLOD-TTA” and “B. Norm VLOD-TTA” use our proposed loss while adapting adapter and batch norm parameters, respectively. Batch norm with our loss reaches almost the same performance as the adapter baseline, which shows that the improvement does not simply come from the additional adapter parameters.
>
> **Why optimize adapters instead of batch norm?** With adapters, the best performance is obtained across datasets with a single learning rate of 5e-3. For batch norm, different learning rates are required to reach the best performance for each dataset, for example, 1e-2 for Watercolor and 3e-2 for ClipArt. In the TTA setting, the target domain is unknown at test-time, so tuning the learning rate per-dataset is impractical. For this reason, we choose to update adapter parameters in our main experiments. We have added a comparison of batch norm versus adapters, including the table below, in Section 4.3 (Ablation Studies) of the revised manuscript.
>
> | Method          |      | Water |       |      | ClipArt |       |      | Comic |       |
> | :-------------- | :--: | :--------: | :---: | :--: | :-----: | :---: | :--: | :---: | :---: |
> |                 |  mAP |    mAP50   | mAP75 |  mAP |  mAP50  | mAP75 |  mAP | mAP50 | mAP75 |
> | Zero-shot       | 26.9 |    47.9    |  25.9 | 24.4 |   40.1  |  26.3 | 17.8 |  29.4 |  18.8 |
> | B.Norm          | 28.4 |    51.3    |  26.5 | 26.7 |   44.1  |  28.1 | 20.6 |  34.5 |  21.7 |
> | Adapters        | 28.3 |    51.5    |  26.7 | 26.9 |   44.1  |  27.8 | 20.8 |  34.7 |  21.7 |
> | **B.Norm VLOD-TTA** | 29.4 |    52.9    |  **28.8** | **28.3** |   45.3  |  **30.0** | 21.3 |  **36.1** |  22.0 |
> | **VLOD-TTA**        | **29.6** |    **53.1**    |  28.7 | 28.1 |   **45.4**  |  29.9 | **21.4** |  **36.1** |  **22.1** |
>
>
>
> > ### **W1.2 Complexity analysis of learnable parameters and memory complexity**
>
> An in-depth complexity analysis of our method and the baselines is provided in Appendix A6. Our method shows a favorable robustness–cost trade-off with consistent gains over the baselines.
>
> **Complexity Comparison with Batch Norm -** Batch norm has 0.021M trainable parameters and an FPS of 21, while adapters have 1.61M trainable parameters and an FPS of 20. In terms of memory, batch norm is lighter and the FPS is almost identical for both. However, batch norm requires fine-tuning the learning rate for each dataset to achieve a high level of performance, which reduces its practical applicability.
>
> > ### **W2.1 Contribution of individual components**
>
> We thank the reviewer for requesting an important ablation. The table below isolates each module. “Adapter” denotes standard entropy minimization with only lightweight adapters updated. “IWE” replaces the entropy loss with IoU-weighted entropy. “VLOD-TTA” combines IWE with image-conditioned prompt selection (IPS). Replacing standard entropy with IWE improves both localization (mAP, mAP75) and classification (mAP50) across Watercolor, ClipArt, and Comic datasets. Adding IPS on top of IWE consistently yields further gains, giving the best overall performance. Each component contributes positively, and the combination achieves the best performance. We have added an ablation on the contribution of individual components in Section 4.3 (Ablation Studies) of the revised manuscript.
>
> | Method          |      | Water |       |      | ClipArt |       |      | Comic |       |
> | :-------------- | :--: | :--------: | :---: | :--: | :-----: | :---: | :--: | :---: | :---: |
> |                 |  mAP |    mAP50   | mAP75 |  mAP |  mAP50  | mAP75 |  mAP | mAP50 | mAP75 |
> | Zero-shot        | 26.9 |    47.9    |  25.9 | 24.4 |   40.1  |  26.3 | 17.8 |  29.4 |  18.8 |
> | Adapter         | 28.3 |    51.5    |  26.7 | 26.9 |   44.1  |  27.8 | 20.8 |  34.7 |  21.7 |
> | IWE             | 29.3 |    52.6    |  28.6 | 27.5 |   44.7  |  29.0 | 21.2 |  35.6 |  22.0 |
> | **VLOD-TTA (IWE + IPS)** | **29.6** |    **53.1**    |  **28.7** | **28.1** |   **45.4**  |  **29.9** | **21.4** |  **36.1** |  **22.1** |

---

> ### Author Response · Authors · 2025-11-21
> **Response to Reviewer vTge [2/2]**
>
> > ### **W2.2, Q1, Q2. How are textual prompts generated?**
>
> We used ChatGPT-5 to generate textual prompts and also tested ChatGPT-4.1, which produced nearly identical results. For each dataset, we asked the model: “Generate 16 prompts for each object category: {*category list*}”. To support reproducibility, we will release step-by-step instructions and a script to generate prompts for any custom dataset. We provide examples of prompts in Appendix A8. We have added more details on prompt generation in Section 4.1 (Implementation Details) of the revised manuscript.
>
> > ### **Q3. Different samples in batches and its impact**
>
> As noted in Section 4.1 (Implementation Details), we use batch size 1 to reflect a realistic online TTA scenario, where images arrive sequentially at deployment. Appendix A9 provides an in-depth analysis of batch size effects. Performance peaks around batch sizes 4–8, showing that our approach is not limited to using a batch size of 1.

---

### Official Review · Reviewer_Eyox · 2025-11-03

**Soundness:** 2
**Presentation:** 3
**Contribution:** 2
**Rating:** 4
**Confidence:** 3

**Summary:**

This paper proposes VLOD-TTA, the first test-time adaptation (TTA) framework tailored specifically for vision-language object detectors (VLODs) such as YOLO-World and Grounding DINO, both of which have demonstrated strong zero-shot generalization. The main technical contributions are twofold: (1) IoU-weighted entropy minimization (IWE), which emphasizes adaptation over spatially coherent clusters of object proposals to address confirmation bias and localization uncertainty, and (2) image-conditioned prompt selection (IPS), which selects and fuses the most relevant textual prompts for each image to improve detection robustness. The approach is empirically validated on a comprehensive benchmark encompassing diverse domain shifts, showing consistent improvements over both zero-shot and existing TTA baselines.

**Strengths:**

1. The paper proposes a well-motivated VLOD-TTA framework, which is interesting and inspiring.

2. Clear motivation and problem setup for TTA in VLODs. Figures 1 and 2 (Pages 2) concretely illustrate failure modes of standard entropy and uniform prompt averaging, and how the proposed IWE and IPS address them.

3. Comprehensive experimental evaluation and Clear presentation and informative figures/tables.

**Weaknesses:**

1. Despite comprehensive benchmarks, the paper mostly highlights consistent positive gains. However, as briefly noted in the conclusion and Section 4.4, IWE can underperform in scenes with numerous small, scattered objects (e.g., Cityscapes); yet the depth of analysis is minimal. It would significantly strengthen the paper to have a more granular breakdown for such problematic cases, including visualizations and quantification of failure cases.

2. As shown in Figure 5, the placement of adapters differs between YOLO-World (vision backbone+neck) and Grounding DINO (text encoder), with minimal theoretical motivation provided. This is largely left to empirical ablation, when some architectural analysis or interpretation might clarify practical trade-offs or inform future practitioners.

3. While overall clarity is reasonable, the related work section (Section 2) needs to be more explicit about methodological distinctions versus related TTA for VLM and OD approaches

**Questions:**

See weakness.

---

> ### Author Response · Authors · 2025-11-21
> **Response to Reviewer Eyox [1/2]**
>
> Thank you for your valuable feedback. Below, we respond to each concern raised and highlight the revisions incorporated into the updated manuscript.
>
> > ### **W1. Breakdown of Cityscapes failure case**
>
> Our method underperforms on Cityscapes. A deeper analysis shows two causes and fixes.
>
> **Overlap between rider and person classes -** Cityscapes has 8 classes, including rider and person. In practice, riders are visually very similar to persons, and the zero-shot model has a strong bias toward the person label. As a result, a rider is often localized correctly but predicted as person. Our IoU-weighted entropy (IWE) tends to push these ambiguous cases further toward person because it sharpens high-IoU clusters. This is still counted as an error under the dataset labels, but it mainly reflects the zero-shot bias toward person rather than a failure to detect riders. This observation is shown in the table below, where we report class-wise AP for person and rider separately. Compared to the zero-shot model, VLOD-TTA reduces the AP for both person and rider.
>
> | Cityscapes                  | Person (AP) | Rider  (AP) |
> | --------------------------- | ----------- | ----------- |
> | Zero-shot                   | 17.8        | 6.3         |
> | VLOD-TTA                | 16.5        | 1.9         |
> | Zero-shot (merged rider)    | 23.2        | -           |
> | **VLOD-TTA (merged rider)** | **24.4**   | -          |
>
> **Experiment -** We merged the rider class into person to create a 7-class annotation set. The zero-shot baseline increases after mapping rider to person (see the table below), and the class-wise AP in the previous table shows that this corresponds to a higher AP for the merged person class. Our VLOD-TTA further improves because IWE no longer distributes probability between rider and person, so entropy minimization focuses on unified person clusters and leads to larger gains.
>
> | Cityscapes         | mAP  | mAP50 | mAP75 |
> | ------------------ | ---- | ----- | ----- |
> | Zero-shot                 | 18.8 | 31.0  | 17.9  |
> | VLOD-TTA                | 19.4 | 31.8  | 18.6  |
> | Zero-shot (merged rider)  | 21.3 | 34.7  | 20.5  |
> | **VLOD-TTA (merged rider)** | **22.5** | **35.9**  | **21.1**  |
>
> **Small Objects -** Cityscapes contains many small objects, with the original resolution being 2048×1024. YOLO-World resizes inputs to 640×640 (by default), which further shrinks distant cars and pedestrians and causes missed detections. At this resolution, many small objects do not generate stable overlapping proposals, so IWE underperforms.
>
> **Experiment -** We evaluate the model with images of 1024×1024 input resolution. The zero-shot baseline increases substantially with the increase in image resolution. VLOD-TTA then improves further, with a modest gain at 640 (+0.6 mAP) but a much larger gain at 1024 (+2.7 mAP). At higher resolution, more small objects produce stable overlapping proposals that IWE can weight and sharpen during adaptation. Small objects are shown to be the main reason for the underperformance.
>
> | Cityscapes     | mAP  | mAP50 | mAP75 |
> | -------------- | ---- | ----- | ----- |
> | Zero-shot (640)            | 18.8 | 31.0  | 17.9  |
> | VLOD-TTA (640)          | 19.4 | 31.8  | 18.6  |
> | Zero-shot (1024)  | 28.5 | 43.9  | 28.3  |
> | **VLOD-TTA (1024)** | **31.2** | **46.6**  | **30.5**  |
>
> **Takeaway -** The underperformance on Cityscapes is due to label overlap and small objects. After addressing both, VLOD-TTA shows larger gains. We have added the visualizations and two ablations for an in-depth analysis of the Cityscapes underperformance in Appendix A2 of the revised manuscript.

---

> ### Author Response · Authors · 2025-11-21
> **Response to Reviewer Eyox [2/2]**
>
> > ### **W2. Justification of placement of adapters in different locations in YOLO-World and Grounding DINO**
>
> We agree with the reviewer and have clarified this in Section 3.4 (Model Update) of the revised manuscript. Vision-language object detectors (VLODs) have two separate modules, a text encoder and an object detector. The information from these two modules is merged to perform object detection. YOLO-World and Grounding DINO architecturally differ in how they merge the information from these two modules.
>
> * **YOLO-World -** YOLO-World treats the detector and the text encoder as isolated blocks. The detector computes bounding-box scores and vision class embeddings, while the text encoder outputs text class embeddings. Final detections are obtained by cosine similarity between vision and text embeddings followed by post-processing. The bounding-box scores and text embeddings do not interact inside the detector. Adapting the text encoder, therefore, mainly affects classification scores (not localization), which leads to limited gains. In fact, the original YOLO-World paper [1] notes that fine-tuning the text encoder during pretraining hurts performance, so our strategy aligns with that finding. We thus adapt only the vision module. We omitted adapters in the detector head to reduce computational complexity. Adapters can be added to the head in addition to the backbone and neck without reducing performance (see Fig. 5 of our main paper).
>
> * **Grounding DINO -** Grounding DINO emphasizes early cross-modal fusion. The Grounding DINO paper [2] states that early “tight modality fusion” is key for vision-language detection. The architecture uses language-guided query selection and a cross-modal decoder, where text embeddings guide detection. Hence, adapting the text encoder can directly improve overall detection (both localization and classification). The original Grounding DINO paper notes that during pretraining, the text encoder is optimized along with the rest of the model. For this reason, we train adapter parameters in the text encoder at test time. The weaker results when optimizing the vision detector are likely due to overfitting. Grounding DINO is much heavier than YOLO-World. YOLO-World-Small has around 13M parameters, while Grounding DINO-Tiny has around 172M. Adding vision-side adapters to the large Grounding DINO detector tended to overfit when optimizing on a single image.
>
> **Conclusion -** Overall, the two detectors differ substantially in terms of the architecture and pretraining, so we place adapters in different locations for each model. Our empirical results in Section 4.3 confirm that adapter placement has a strong impact on performance, yet TTA for VLODs is still largely unexplored and a generally effective adapter placement strategy remains an open question.
>
>
> > ### **W3. Distinction from VLM TTA and TTA of ODs**
>
> Below, we distinguish our VLOD-TTA from prior VLM-TTA and OD-TTA and have incorporated the text into Section 2 of the revised manuscript.
>
> * **VLM TTA -** Prior TTA methods for VLMs focus only on the classification task. Adaptation is performed at the image level and does not involve region proposals or a detection head, so localization is not addressed. In contrast, our VLOD-TTA adapts at the proposal level for open-vocabulary detection, improving both localization and classification.
> * **TTA of ODs -** State-of-the-art TTA methods for ODs target closed-set ODs with a fixed, vision-only label space. Models are first trained on a source domain similar to the target domain. In addition, many methods require access to source data statistics during adaptation, which limits their applicability to VLODs. In contrast, our VLOD-TTA method adapts VLODs in an open-vocabulary label space, leveraging both visual and textual cues.
>
> ### **References**
>
> *[1] Cheng, T., Song, L., Ge, Y., Liu, W., Wang, X., & Shan, Y. (2024). Yolo-world: Real-time open-vocabulary object detection. In Proceedings of the IEEE/CVF conference on computer vision and pattern recognition (pp. 16901-16911).*
>
> *[2] Liu, S., Zeng, Z., Ren, T., Li, F., Zhang, H., Yang, J., ... & Zhang, L. (2024, September). Grounding dino: Marrying dino with grounded pre-training for open-set object detection. In European conference on computer vision (pp. 38-55). Cham: Springer Nature Switzerland.*

---

### Author Response · Authors · 2025-11-21
**Global Response**

We thank all reviewers for their insightful comments and suggestions, and have revised the manuscript to address their concerns. Our modifications are highlighted in blue in the revised PDF. The major revisions include an ablation study on batch norm parameters, a theoretical justification of our detector-specific adapter placement, an analysis of the contribution of each component, a Cityscapes failure case study, and an experiment combining IoU weighting with a pseudo-labeling scheme. We also appreciate the positive feedback on our benchmarking. We are encouraged that our work was found to be well-motivated (Reviewer sLD7), the idea original (Reviewer vTge), the problem relevant (Reviewer YXiS), and the presentation easy to follow (Reviewer 2uSN). Below, we respond to each reviewer’s comments in detail and are happy to clarify any remaining questions.

---

### Author Response · Authors · 2025-12-04
**Summary of the Rebuttal**

Dear Area Chair,

We sincerely thank you for your time and dedication. We also appreciate the constructive feedback from all reviewers, which has significantly strengthened our work. The manuscript has been revised to address the reviewers' concerns. All modifications are highlighted in blue in the revised version. To facilitate reading our rebuttal, we provide a summary of the existing reviews and responses.

**1. Primary Strengths:**

We appreciate the following positive recognition of our work by reviewers:
* The work addresses an underexplored problem of test-time adaptation (TTA) for vision-language object detectors (VLODs) with a well-motivated VLOD-TTA framework. *[YXiS, Eyox]*
* The problem setup and motivation are clear, and the idea is original and interesting. *[vTge, Eyox]*
* The empirical benchmark is comprehensive, showing consistent gains across multiple datasets and two state-of-the-art models. *[YXiS, vTge, Eyox, 2uSN]*

**2. Primary Revisions:**

The reviewers’ suggestions for improving the manuscript are summarized below:
* Can batch normalization parameters be trained instead of adapters? *[vTge W1.1, YXiS W3]*
    * We conducted experiments where batch normalization parameters are adapted, showing results equivalent to adapters. However, adapters reach their best performance across datasets with a single learning rate, whereas batch normalization requires dataset-specific tuning. This is impractical for TTA, since the target domain is unknown at test time, so we optimize adapter parameters in our main experiments. (Section 4.3)
* Provide a justification of architecture dependent adapter placement *[Eyox W2, YXiS W1]*
    * We provide a theoretical justification of our adapter placement and show that the different adapter locations are due to the very different architectures and pre-training strategies of each model, rather than a lack of generality of our method. As the first work to study TTA for VLODs, this analysis also establishes a strong benchmark for future methods. (Section 3.4)
* Provide an in-depth analysis of Cityscapes failure case *[Eyox W1, 2uSN W1]*
    * We performed a deeper analysis and identified two main causes of underperformance, namely label overlap between person and rider and many small objects. To overcome this, we merge rider into person and evaluate at a higher input resolution, which leads to larger gains for VLOD-TTA on the dataset. (Appendix A2)
* Limited scope of our modules *[YXiS W4]*
    * To demonstrate the applicability of our IoU-weighting beyond entropy minimization, we integrate it into a standard pseudo-label-based TTA scheme for ODs and observe consistent improvements, highlighting the broader applicability of our design. (Appendix A3)

**3. Further Revisions:**

In addition to the points above, we made several further revisions to improve clarity:
* The contribution of each individual component of our method is analyzed through an ablation study *[vTge W2.1]*. (Section 4.3)
* The effectiveness of IPS in specialized domains is evaluated through an ablation *[2uSN W2.2]*. (Appendix A13)
* The related work section is refined to better distinguish TTA for vision–language models from TTA for conventional ODs *[Eyox W3]*. (Section 2)
* Additional details are provided on how textual prompts are generated *[vTge W2.2]*. (Section 4.1)
* The training protocol used in the model update step is described in more detail *[YXiS W2]*. (Section 3.4)

During the rebuttal, the questions raised by the reviewers primarily focused on additional ablation studies and deeper insights into the proposed method. We addressed these points with new ablations and clarifications, which further strengthen both the empirical and conceptual support for VLOD-TTA.

---

### Meta-Review · Area_Chair_9GUq · 2026-01-09

**Summary:**

The paper proposes a method for test-time adaptation of open-vocabulary detectors. It involves (1) IoU-weighted entropy objective that helps the detector focus on clusters of candidates as opposed to spurious detections; (2) image-conditioned prompt selection, intelligently selecting prompts per class based on the image. The method consistently matches or outperforms comparable adaptation methods across several tasks and 2 models

Based on the reviews, the authors’ rebuttal, and the paper itself, the main strengths and weaknesses are as follows.

Pros:
1. A practical problem statement and a reasonable motivation.
2. Good presentation, including a good discussion of downsides
3. Reasonable and clear approach
4. Good performance across 2 models and several tasks.
5. The rebuttal addressed some issues raised by the reviewers: adapters vs fine-tuning normalization layer parameters, architecture-dependent aspects of the method, cityscapes failure case.

Cons:
1. The contributions are engineering-flavored: not fundamental ML improvement or brave new approach, but pragmatic fixes of issues for a specific class of models. Pragmatically fixing issues is not a bad thing per se, but for publication such engineering-heavy works are expected to be very strong in other respects
2. Performance improvements are moderate: around 0.5-3%, on average perhaps about 1-1.5%. This is again not bad, but also not huge.

Overall, the paper is quite borderline. The reviewers were originally not very positive, but I believe the authors did a good job addressing their concerns. There are many good things to be said about the paper, but in the end the technical contributions are engineering-heavy and specialized, while the performance improvement is not dramatic. Therefore at this point I lean towards rejection, but very much encourage the authors to improve the paper and resubmit to a different venue.

**Reviewer Concerns:**

- Adapters vs fine-tuning parameters of normalization layers -> the authors provided the results in the rebuttal and have shown that this doesn’t make much difference
- Architecture-dependent aspects of the method -> The authors provided a reasonable explanation of why this is necessary
- Cityscaped failure -> again, the authors provided a good analysis of why this happens and a reasonable explanation

**Reviewer Scores:**

I think 2-3 reviewers might have raised their scores by 1 point, since the authors did a good job addressing the reviewers’ concerns.

---

### Decision · Program_Chairs · 2026-01-26

Reject